# $f$-MICL: Understanding and Generalizing InfoNCE-based Contrastive Learning

**Yiwei Lu** [*]                                               *yiwei.lu@uwaterloo.ca*
*School of Computer Science*
*University of Waterloo*
*Vector Institute*

**Guojun Zhang** [*]                                           *guojun.zhang@huawei.com*
*Huawei Noah's Ark Lab*

**Sun Sun**                                                    *sun.sun@nrc-cnrc.gc.ca*
*School of Computer Science*
*University of Waterloo*
*National Research Council Canada*

**Hongyu Guo**                                                 *hongyu.guo@uottawa.ca*
*National Research Council Canada*
*University of Ottawa*

**Yaoliang Yu**                                                *yaoliang.yu@uwaterloo.ca*
*School of Computer Science*
*University of Waterloo*
*Vector Institute*

**Reviewed on OpenReview:** *https://openreview.net/forum?id=ZDO3VUZmRx*

## Abstract

In self-supervised contrastive learning, a widely-adopted objective function is InfoNCE, which uses the heuristic cosine similarity for the representation comparison, and is closely related to maximizing the Kullback–Leibler (KL)-based mutual information. In this paper, we aim at answering two intriguing questions: (1) Can we go beyond the KL-based objective? (2) Besides the popular cosine similarity, can we design a better similarity function? We provide answers to both questions by generalizing the KL-based mutual information to the $f$-**M**utual **I**nformation in **C**ontrastive **L**earning ($f$-MICL) using the $f$-divergences. To answer the first question, we provide a wide range of $f$-MICL objectives which share the nice properties of InfoNCE (e.g., alignment and uniformity), and meanwhile result in similar or even superior performance. For the second question, assuming that the joint feature distribution is proportional to the Gaussian kernel, we derive an $f$-**Gaussian similarity** with better interpretability and empirical performance. Finally, we identify close relationships between the $f$-MICL objective and several popular InfoNCE-based objectives. Using benchmark tasks from both vision and natural language, we empirically evaluate $f$-MICL with different $f$-divergences on various architectures (SimCLR, MoCo, and MoCo v3) and datasets. We observe that $f$-MICL generally outperforms the benchmarks and the best-performing $f$-divergence is task and dataset dependent.

---

[*]Equal contribution

# 1  Introduction

Recent advances in self-supervised learning aim at learning similar representations from different augmented views of the same data sample. However, naively implementing this idea would easily make representations converge to some trivial constant (i.e., feature collapse) in practice. To address this problem, researchers propose new algorithms either from the *model architecture* perspective or the *training objective* perspective. The former method (e.g., Grill et al. (2020); Chen & He (2021); Zhang et al. (2021)) applies techniques such as stop gradient or predictor module to create asymmetry in networks, while the latter method encourages the contrastiveness between similar (positive) and dissimilar (negative) sample pairs through the objective design.

In this paper, we intend to deepen our understanding of contrastive learning by generalizing the current objective design. To achieve self-supervised contrastive learning, existing objectives are proposed from different perspectives such as the mutual information (e.g., InfoNCE (Wu et al., 2018; van den Oord et al., 2018; Chen et al., 2020; Hénaff et al., 2020; He et al., 2020) ), the information redundancy (e.g., Barlow Twins (Zbontar et al., 2021)), and the regularization (e.g., VICReg (Bardes et al., 2021)). In particular, the InfoNCE objective is widely used, which aims to maximize the probability of picking a similar sample pair among a batch of sample pairs, and can be interpreted as a lower bound of the mutual information (MI) between two views of samples (van den Oord et al., 2018; Bachman et al., 2019; Tian et al., 2020a; Tschannen et al., 2020). This is consistent with the well-known "InfoMax principle" (Linsker, 1988). To measure the similarity between sample pairs, cosine similarity is usually adopted.

To attain the aforementioned goals of better understanding and generalizing contrastive learning, we here focus on the widely-adopted InfoNCE objective, and aim at two questions regarding it:

*(1) MI is essentially the Kullback–Leibler (KL) divergence between the joint distribution and the product of the marginal distributions. Is this KL-based objective optimal? If not, can we go beyond the KL-based objective?*

*(2) Besides the commonly used cosine similarity for measuring the distance between samples, can we provide a better similarity function with a theoretical basis?*

To answer the above two questions, we generalize the KL-based mutual information to the broader $f$-divergence family (Ali & Silvey, 1966; Csiszár, 1967), and propose the benchmark of $f$-mutual information in contrastive learning ($f$-MICL). By searching through a wide range of $f$-divergences, we observe that the KL divergence is not always the best, and several other $f$-divergences in fact show similar or even superior performance in practice.

For the second question, while it is challenging to provide an answer based on the InfoNCE objective, it is possible to derive a proper similarity function under the $f$-MICL framework. By assuming that the joint feature distribution is proportional to the popularly-adopted Gaussian kernel, we propose a novel $f$-*Gaussian similarity* function that enjoys better empirical performance.

Finally, we show the generalization of $f$-MICL by drawing connections between the $f$-MICL objective and several popular InfoNCE-based objectives (e.g., SimCLR(Chen et al., 2020), MoCo(He et al., 2020), and Alignment and Uniformity (AU) (Wang & Isola, 2020)). We identify that those objectives are closely related to $f$-MICL: Alignment and Uniformity (AU) (Wang & Isola, 2020) can be treated as a special case, and SimCLR (Chen et al., 2020) and MoCo (He et al., 2020) are upper bounds for a transformed $f$-MICL. These results provide a different angle to better understand InfoNCE. Moreover, we show both theoretically and empirically that nice properties of InfoNCE (e.g., alignment and uniformity (Wang & Isola, 2020)) can be naturally extended to $f$-MICL.

We summarize our main contributions as follows:

- Motivated by InfoNCE, we propose a general framework for contrastive learning by extending the MI to the general $f$-MI, which provides a wide range of objective choices.

- Instead of using heuristic similarity functions, we provide a novel similarity function, called $f$-*Gaussian similarity*, based on the convex conjugate and an assumption on the joint feature distribution.
- We identify close relationships between our $f$-MICL objective and several InfoNCE-based contrastive learning objectives.
- Empirically, we show that $f$-MICL achieves notable improvement over benchmarks on various datasets, and the best-performing $f$-divergence depends on the specific task and dataset. In addition, our proposed $f$-Gaussian similarity consistently outperforms the cosine similarity.

## 2  $f$-Mutual Information

To provide answers to the above two questions regarding the InfoNCE objective, we first extend the KL-based mutual information to the more general $f$-mutual information. The definition of the $f$-mutual information ($f$-MI) is as follows:

**Definition 1** ($f$-mutual information, Csiszár 1967). *Consider a pair of random variables $(X, Y)$ with joint density function $p(x, y)$ and marginal densities $p(x)$ and $p(y)$. The $f$-mutual information $I_f$ between $X$ and $Y$ is defined as*

$$I_f(X;Y) := \int f\left(\frac{p(x,y)}{p(x)p(y)}\right) p(x)p(y) \cdot \mathrm{d}\lambda(x,y), \tag{1}$$

*where $f : \mathbb{R}_+ \to \mathbb{R}$ is (closed) convex with $f(1) = 0$, and $\lambda$ is a dominating measure (e.g., Lebesgue).*

Note that the $f$-MI is essentially the $f$-divergence between the joint distribution and the product of marginal distributions. It is well-known that the $f$-MI is non-negative and symmetric. Moreover, provided that $f$ is strictly convex, $I_f(X;Y) = 0$ iff $X$ and $Y$ are independent (Ali & Silvey, 1966).

Since it is challenging to provide an accurate estimation of the $f$-divergences in high dimensions, Nguyen et al. (2010) used the convex conjugate as a lower bound for the $f$-divergences. With this result we can lower bound $I_f(X;Y)$ as follows:

$$I_f(X;Y) \geq \sup_{s \in \mathcal{F}} \left( \mathbb{E}_{(X,Y) \sim p(x,y)} s(X,Y) - \mathbb{E}_{(X,Y) \sim p(x)p(y)} f^* \circ s(X,Y) \right), \tag{2}$$

where $p(x, y)$ denotes the joint density, $p(x)p(y)$ stands for the product of marginal densities, and the symbol $\circ$ denotes function composition. The function $f^*(t) := \sup_{x \in \mathbb{R}_+}(xt - f(x))$ is known as the convex conjugate[1] of $f$ and is *monotonically increasing*, and $s(\cdot)$ belongs to $\mathcal{F}$, a class of functions on $(x, y)$ that we can parameterize. Using results in Nguyen et al. (2010), one can show that eq. (2) is equal to $I_f(X;Y)$ if there exists $s_\star \in \mathcal{F}$ such that for any $(x, y) \in \mathrm{supp}(p(x)) \times \mathrm{supp}(p(y))$, where $\mathrm{supp}(\cdot)$ denotes the support of a distribution, we have:

$$s_\star(x,y) = f'\left(\frac{p(x,y)}{p(x)p(y)}\right). \tag{3}$$

In other words, plugging the optimal $s_\star(x, y)$ into eq. (2) we obtain equality. In Table 1 we list common choices of $f$-divergences, their conjugates, and the derivatives. We also include the composition $f^* \circ f'$ for later purposes (see Theorem 4 below).

## 3  $f$-MICL

With the introduction of the more general $f$-MI we now proceed to the design of the objective and similarity function. Then we will analyze the property of the proposed framework and compare it with some existing InfoNCE-based benchmarks.

---

[1]More precisely, this is the monotone convex conjugate since we restrict the domain of $f$ to $\mathbb{R}_+$.

Table 1: Common choices of $f$-divergences. KL: Kullback–Leibler; JS: Jensen–Shannon; SH: Squared Hellinger; VLC: Vincze–Le Cam (Le Cam, 2012). For JS, we define $\varphi(u) = -(u+1)\log\frac{1+u}{2} + u\log u$. The Tsallis-$\alpha$ divergence is defined in Tsallis (1988). See Appendix A.1 for more details.

| **Divergence** | $f(u)$ | $f^*(t)$ | $f'(u)$ | $f^* \circ f'(u)$ |
|---|---|---|---|---|
| KL | $u\log u$ | $\exp(t-1)$ | $\log u + 1$ | $u$ |
| JS | $\varphi(u)$ | $-\log(2-e^t)$ | $\log 2 + \log\frac{u}{1+u}$ | $-\log\frac{2}{1+u}$ |
| Pearson $\chi^2$ | $(u-1)^2$ | $t^2/4 + t$ | $2(u-1)$ | $u^2 - 1$ |
| SH | $(\sqrt{u}-1)^2$ | $\frac{t}{1-t}$ | $1 - u^{-1/2}$ | $u^{1/2} - 1$ |
| Tsallis-$\alpha$ | $\frac{u^\alpha}{\alpha-1}$ | $\left(\frac{\alpha-1}{\alpha}t\right)^{\frac{\alpha}{\alpha-1}}$ | $\frac{\alpha}{\alpha-1}u^{\alpha-1}$ | $u^\alpha$ |
| VLC | $\frac{(u-1)^2}{u+1}$ | $4 - t - 4\sqrt{1-t}$ | $1 - \frac{4}{(u+1)^2}$ | $3 - \frac{4}{u+1}$ |

### 3.1 $f$-MICL objective

Contrastive learning is a popular *self-supervised* method for representation learning. In contrastive learning, we expect similar sample pairs to be close to each other in the embedding space, while dissimilar pairs to be far apart. Based on the $f$-MI introduced in § 2, we propose a general framework for contrastive learning, coined as $f$-MICL.

We denote $g : \mathcal{X} \to \mathbb{S}^{d-1}$ as the feature encoder (usually constructed by a neural network) from the input space $\mathcal{X}$ to the hypersphere, and we use the shorthands $x^g := g(x)$ and $y^g := g(y)$ to represent the feature embeddings. The notation $p_d$ stands for the data distribution, and $p_\times := p_d \otimes p_d$ means its self product (product of marginals, e.g., pairs of images). We denote $p_+$ as the distribution of *positive pairs*, *i.e.*, two samples with similar feature embeddings (joint distribution, e.g., the same image with different data augmentation). Using the lower bound of the $f$-MI in eq. (2), we have the general $f$-MICL objective as follows:

$$\sup_{s \in \mathcal{F}} \ \mathbb{E}_{(x,y)\sim p_+} s(x^g, y^g) - \mathbb{E}_{(x,y)\sim p_\times} f^* \circ s(x^g, y^g), \tag{4}$$

where $s(x^g, y^g)$ can be understood as the similarity measurement between two feature embeddings in the context of contrastive learning. Essentially, we are studying the variational lower bound eq. (2) in the *feature space*, with the feature embeddings learnable. We can treat the first term as the similarity score between *positive pairs* with similar feature embeddings, and the second term as the similarity score between two random samples, a.k.a. *negative pairs*. As $f^*$ is an increasing function, maximizing the $f$-MI is equivalent to simultaneously maximizing the similarity between positive pairs and minimizing the similarity between negative pairs.

With eq. (4) we have answered the first question: there are a spectrum of $f$-MICL objectives that can be applied in contrastive learning by using different $f$ functions. We will discuss how to choose the best $f$ empirically in §4.

### 3.2 $f$-Gaussian similarity

Previous works in constrastive learning usually adopt some heuristic similarity function such as the cosine similarity function $s(x^g, y^g) = x^g \cdot y^g$. Although it shows promising performance in practice, our second question is that, can we provide a better similarity function than the popular cosine similarity?

Note that eq. (3) is a natural choice of $s(\cdot)$ from the perspective of deriving the $f$-MI. In the context of contrastive learning, by denoting the density functions of the marginal feature distributions as $p_g(x^g)$ and $p_g(y^g)$, and the density of the joint feature distribution as $p_g(x^g, y^g)$, from eq. (3) we have an optimal similarity function as follows:

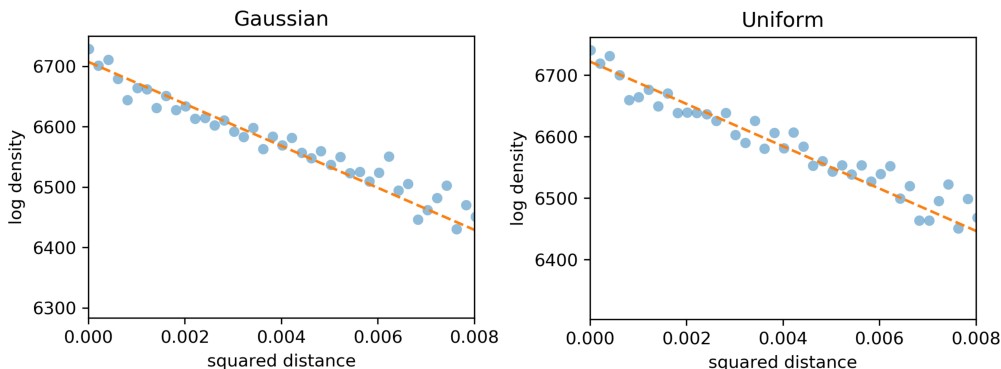

Figure 1: Experiment for verifying Assumption 3. Here we draw the relation between the squared distances $\|x^g - y^g\|^2$ and the averaged log likelihood $\log p_g$, with $\log p_g$ estimated by the flow model RealNVP (Dinh et al., 2017). (**left**) Gaussian prior; (**right**) Uniform prior. The features are learned by SimCLR trained on CIFAR-10. See more details in Appendix D.3.

**Lemma 2** (*e.g.*, Nguyen et al. 2010, Lemma 1)**.** *Suppose $f$ is differentiable, and the embedding function $g$ is fixed. The following similarity function $s_\star$ maximizes eq. (4):*

$$s_\star(x^g, y^g) = f'\left(\frac{p_g(x^g, y^g)}{p_g(x^g)p_g(y^g)}\right). \tag{5}$$

Obviously, the optimal $s_\star$ in fact induces the $f$-MI on the feature space, which is a low bound of the original $f$-MI. Although eq. (5) provides an optimal similarity function, it nevertheless depends on the unknown density functions. How can we implement eq. (5) in practice? Among various known density functions, it is natural to choose a typical kernel function for structured data for validation (Balcan et al., 2008; Powell, 1987; Murphy, 2012), e.g., the Gaussian kernel.

**Assumption 3** (**Gaussian kernel**)**.** *The joint feature density (wrt the uniform distribution over the hypersphere $\mathbb{S}^{d-1}$) is proportional to a Gaussian kernel, namely*

$$p_g(x^g, y^g) \propto G_\sigma(\|x^g - y^g\|^2) = \mu \exp\left(-\frac{\|x^g - y^g\|^2}{2\sigma^2}\right),$$

*where $\mu := \exp(\frac{1}{\sigma^2})\frac{C}{c^2}$ is a constant that we determine below.*

Since $x^g, y^g \in \mathbb{S}^{d-1}$ have unit (Euclidean) norm, we have

$$p_g(x^g, y^g) \propto \exp(\tfrac{x^g \cdot y^g}{\sigma^2}), \tag{6}$$

which belongs to the von Mises-Fisher bivariate distribution (Mardia, 1975, Eq. 2.11). It is clear that the marginals of $p_g$ are uniform. Indeed, for any orthogonal matrix $Q$, with $\frac{1}{C} := \mathbb{E}_{(x^g, y^g) \sim \mathbb{S}^{d-1} \times \mathbb{S}^{d-1}} \exp(\tfrac{x^g \cdot y^g}{\sigma^2})$ we have

$$p_g(Qx^g) = \mathbb{E}_{y^g \sim \mathbb{S}^{d-1}} C \exp(\tfrac{Qx^g \cdot y^g}{\sigma^2}) = \mathbb{E}_{y^g \sim \mathbb{S}^{d-1}} C \exp(\tfrac{x^g \cdot Q^\top y^g}{\sigma^2}) = p_g(x^g) =: c, \tag{7}$$

where we have used the fact that the only invariant distribution on $\mathbb{S}^{d-1}$ wrt the orthogonal group is the uniform distribution. Similarly, we have $p_g(y^g) \equiv c$ (where $c$ is the reciprocal of the surface area of the hypersphere $\mathbb{S}^{d-1}$). The distribution (6) has a nice interpretation[2] in terms of maximum entropy (Mardia, 1975), and admits the factorization

$$p_g(x^g, y^g) = p_g(x^g) \cdot p_g(y^g|x^g) = p_g(y^g) \cdot p_g(x^g|y^g), \tag{8}$$

---

[2]As suggested by the action editor, we may also interpret the distribution (6) as a "copula," i.e., a joint density on $\mathbb{S}^{d-1} \times \mathbb{S}^{d-1}$ with uniform marginals. Note that the conventional notion of copula replaces the hypersphere $\mathbb{S}^{d-1}$ with the unit interval $[0, 1]$. More generally, we could consider the "copula" $p_g(x^g, y^g) \propto h(x^g \cdot y^g)$ for an increasing function $h$, to capture other types of correlation between the two views $x^g$ and $y^g$.

where the marginals $p_g(x^g)$ and $p_g(y^g)$ are uniform while the conditionals $p_g(y^g|x^g)$ and $p_g(x^g|y^g)$ again belong to the von Mises-Fisher distribution.

Combining eq. (5) and Assumption 3, we can write the similarity function with the Gaussian kernel as follows:

$$s_f(x^g, y^g) = f' \circ G_\sigma(\|x^g - y^g\|^2). \tag{9}$$

As noted above, the product of marginals $p_g(x^g)p_g(y^g)$ is a constant, which has been absorbed into our definition of $G_\sigma$, see $\mu$ in Assumption 3. We observe that $s_f(\cdot)$ depends on the choice of $f$ as well, thus we call it $f$-**Gaussian similarity**. As a result, we have provided a new way to design the similarity function, again from the $f$-MI perspective.

**Verifying Assumption 3:** One may question that Assumption 3 can be too strong for practical usage. For example, replacing the Gaussian kernel $G_\sigma$ with any other decreasing function would also provide a valid assumption. However, we found that among several popular choices only the Gaussian kernel works well in practice. Also, we can empirically verify that Assumption 3 approximately holds. To this end, it is sufficient to check whether the log density, $i.e.$, $\log p_g(x^g, y^g)$, is linear with the distance between each positive pair, $i.e.$, $\|x^g - y^g\|^2$. In Figure 1, we use the flow-based model RealNVP (Dinh et al., 2017)[3] to estimate the log density with a Gaussian prior and a uniform prior, and learn the feature encoder $g$ from SimCLR (Chen et al., 2020). We observe that the linear relationship approximately holds for CIFAR-10 [4].

We will empirically compare our $f$-Gaussian similarity with the cosine similarity in §4.

### 3.3 Implementation

With our designed $f$-Gaussian similarity $s_f$ we now have an implementable $f$-MICL objective in eq. (4). Bringing the $f$-Gaussian Similarity $s_f$ in eq. (9) into our objective eq. (4) we have a specific $f$-MICL objective:

$$\mathbb{E}_{(x,y) \sim p_+} s_f(x^g, y^g) - \mathbb{E}_{(x,y) \sim p_\times} f^* \circ s_f(x^g, y^g). \tag{10}$$

Given a batch of $N$ samples, its empirical estimation is as follows:

$$\frac{1}{N} \sum_{i=1}^{N} s_f(x_i^g, y_i^g) - \frac{1}{N(N-1)} \sum_{i \neq j} f^* \circ s_f(x_i^g, x_j^g), \tag{11}$$

where $x_i$ and $y_i$ are two types of data augmentation of the $i$-th sample, and $x_i$ and $x_j$ are different samples with independently sampled data augmentations.

With the $f$-MICL objective in eq. (11) we propose our algorithm for contrastive learning in Algorithm 1. To balance the two terms in our objective, we additionally include a weighting parameter $\alpha$ in front of the second term (which also absorbs the parameter $\mu$ in $G_\sigma$). This change can still be incorporated within our $f$-MICL framework, as we show in Appendix A.2. Figure 2 gives a high-level summary of our $f$-MICL framework. Given a batch of samples ($e.g.$, images), we generate *positive pairs* via data augmentation and *negative pairs* using other augmented samples in the same batch. This sampling method follows SimCLR (Chen et al., 2020).

### 3.4 $f$-**MICL family**

In this section, we will deepen the understanding of $f$-MICL by drawing connections with some popular constrastive learning methods.

**Connection with InfoNCE:** Firstly, we show that InfoNCE is an upper bound of $f$-MICL. Recall our $f$-MICL objective in eq. (4), and the popular InfoNCE objective $\mathcal{L}_{\text{InfoNCE}}$ as follows (here we take the

---

[3]RealNVP applies real-valued non-volume preserving transformation for log-likelihood computation.

[4]The linear relationship in Figure 1 might also depend on the data, i.e., the CIFAR-10 dataset here. In practice, other customized datasets might require additional verification.

---

**Algorithm 1:** $f$-MICL

---

**Input:** batch size $N$, function $f$, weighting parameter $\alpha$, constant $\mu$ (in $G_\sigma$), variance $\sigma^2$

**1 for** *each sampled mini-batch* $\{z_i\}_{i=1}^N$ **do**

**2**      **for** $k$ *in* $1, \ldots, N$ **do**

**3**          randomly sample two augmentation functions $t_1$ and $t_2$

**4**          $y_k \leftarrow t_1(z_k)$, $x_k \leftarrow t_2(z_k)$

**5**      define $s_f(x^g, y^g) = f' \circ G_\sigma(\|x^g - y^g\|^2)$

**6**      compute $-\mathcal{L}$ as

$$\frac{1}{N} \sum_{i=1}^N s_f(x_i^g, y_i^g) - \frac{\alpha}{N(N-1)} \sum_{i \neq j} f^* \circ s_f(x_i^g, x_j^g)$$

**7**      update $g$ by minimizing $\mathcal{L}$

---

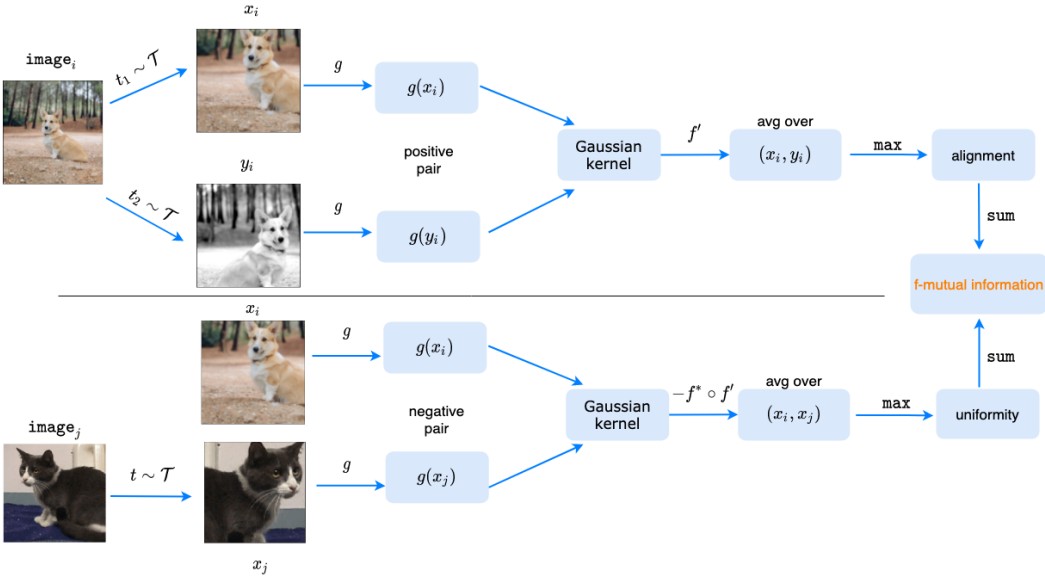

Figure 2: Network architecture of $f$-MICL. $\texttt{image}_i$: the $i^{\text{th}}$ image in the current batch; $f$: the function used in the $f$-mutual information (§2); $g$: feature embedding; $t$, $t_1$, $t_2$: augmentation functions drawn from the same family $\mathcal{T}$ of augmentations; $f'$: the derivative; $f^*$: the Fenchel conjugate. The symbol $\circ$ denotes the function composition. The sum of the two terms gives the variational lower bound of $f$-mutual information. $x_i$ and $y_i$ are two types of data augmentation of the $i$-th sample, and $x_i$ and $x_j$ are different samples with independently sampled data augmentations. $\texttt{max}$ stands for maximization. See eq. (11) for more details.

maximization) (van den Oord et al., 2018):

$$\mathbb{E}_{(x,y) \sim p_+} s(x^g, y^g) - \mathbb{E}_{x \sim p_d} \log \mathbb{E}_{y \sim p_d} \exp(s(x^g, y^g)). \tag{12}$$

Consider that we perform a Donsker-Varadhan (DV) shift transformation $v$ (Donsker & Varadhan, 1975; Tsai et al., 2021) from eq. (4) such that by taking the maximum over the transformation we have:

$$\sup_{v \in \mathbb{R}} \left( \mathbb{E}_{(x,y) \sim p_+} s(x^g, y^g) - v - \mathbb{E}_{(x,y) \sim p_\times} f^* \circ \left( s(x^g, y^g) - v \right) \right). \tag{13}$$

In practice, such a shift transformation can be approximated by a scaling factor ($\alpha$ in Algorithm 1) such that eq. (4) and eq. (13) are equivalent. Given that $f$ is the KL divergence, thus $f(u) = u \log u$ and

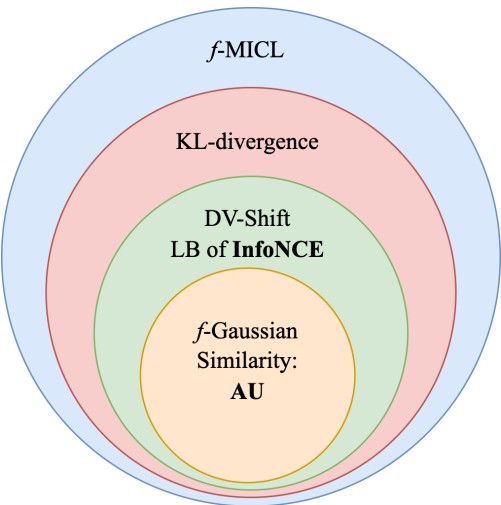

Figure 3: $f$-MICL generalizes InfoNCE-based objectives.

$f^*(t) = \exp(t - 1)$ from Table 1, the maximizer of $v$ in eq. (13) occurs at $v_\star = \log(\mathbb{E}_{(x,y)\sim p_\times} s(x^g, y^g)) - 1)$. With $v_\star$, eq. (13) can be written as follows:

$$\mathbb{E}_{(x,y)\sim p_+} s(x^g, y^g) - \log \mathbb{E}_{(x,y)\sim p_\times} \exp(s(x^g, y^g)). \tag{14}$$

According to Jensen's inequality we have

$$\mathbb{E}_{x\sim p(\cdot)} \log \mathbb{E}_{y\sim p(\cdot)} \exp(s(x^g, y^g)) \leq \log \mathbb{E}_{(x,y)\sim p_\times} \exp(s(x^g, y^g)). \tag{15}$$

Therefore,

$$\mathbb{E}_{(x,y)\sim p_+} s(x^g, y^g) - \log \mathbb{E}_{(x,y)\sim p_\times} \exp(s(x^g, y^g)) \leq \mathcal{L}_{\text{InfoNCE}}. \tag{16}$$

The above transformation shows that the InfoNCE loss is an upper bound of the $f$-MICL objective. In other words, maximizing $f$-MICL can potentially increase the InfoNCE objective.

**Connection with Alignment and Uniformity (AU) (Wang & Isola, 2020):** We further show that the Alignment and Uniformity (AU) loss is a special cases of $f$-MICL. Wang & Isola (2020) shows that InfoNCE approximately aligns positive feature embeddings while encouraging uniformly distributed negative ones. Wang & Isola (2020) further proposes a new objective which quantifies such properties. Here we show that this new objective is essentially a subclass of the InfoNCE loss under the $f$-MICL framework. Concretely, applying the $f$-Gaussian similarity function for the KL divergence, we have $f'(u) = \log u + 1$ from Table 1 and thus $s_f(x^g, y^g) = -\|x^g - y^g\|^2$. Using $s_f(x^g, y^g)$ in eq. (14) we can recover the AU objective:

$$-\mathbb{E}_{(x,y)\sim p_+} \|x^g - y^g\|^2 - \log \mathbb{E}_{(x,y)\sim p_\times} \left[\exp(-\|x^g - y^g\|^2)\right]. \tag{17}$$

Note that for KL, this is equivalent to the cosine similarity with a scaling factor: $-\|x^g - y^g\|^2 = 2x^g \cdot y^g - 2$.

**Connection with the Spectral Contrastive Loss:** Here we show that $f$-MICL objective is closely related to the Spectral Contrastive Loss (HaoChen et al., 2021). Recall our objective:

$$\mathbb{E}_{(x,y)\sim p_+} s(x^g, y^g) - \mathbb{E}_{(x,y)\sim p_\times} f^* \circ s(x^g, y^g), \tag{18}$$

where $s_f(x^g, y^g) = f' \circ G_\sigma(\|x^g - y^g\|^2)$. If we choose the Pearson $\chi^2$ divergence, where $f(u) = (u-1)^2$, $f'(u) = 2(u-1)$, $f^* \circ f'(u) = u^2 - 1$, we have our $\chi^2$-MICL objective:

$$2\mathbb{E}_{(x,y)\sim p_+} G_\sigma(\|x^g - y^g\|^2) - \mathbb{E}_{(x,y)\sim p_\times} G_\sigma(\|x^g - y^g\|^2)^2 - 3. \tag{19}$$

This recovers the spectral contrastive loss exactly if we choose the proper hyperparameter and apply the cosine similarity instead. Thus we generalize the spectral contrastive loss as a special case of $\chi^2$-MICL.

**More on AU:** Finally, based on our objective in eq. (10) we will show that the alignment and uniformity (AU) property of InfoNCE also extends to the general $f$-MICL family: (1) Alignment: In the ideal case, maximizing the first term of eq. (10) would yield $x^g = y^g$ for all $(x, y) \sim p_+$, *i.e.*, similar sample pairs should have aligned representations. Note that the derivative $f'$ is increasing since $f$ is convex. (2) Uniformity: We demonstrate the uniformity property by minimizing the second term of eq. (10), or more rigorously and realistically, its empirical version in eq. (11).

**Theorem 4 (Uniformity).** *Suppose that the batch size $N$ satisfies $2 \leq N \leq d+1$, with $d$ the dimension of the feature space. If the real function*

$$h(t) = f^* \circ f' \circ G_\sigma(t) \text{ is strictly convex on } [0, 4], \tag{20}$$

*then all minimizers of the second term of eq. (11), i.e., $\sum_{i \neq j} f^* \circ s_f(x_i^g, x_j^g)$, satisfy the following condition: the feature representations of all samples are distributed uniformly on the unit hypersphere $\mathbb{S}^{d-1}$.*

In Theorem 4, the assumption $N \leq d+1$ is always satisfied in our experiments in §4. For instance, on CIFAR-10 we chose $N = d = 512$. Also, we claim that the samples are "distributed uniformly" if the feature vectors form a regular simplex, and thus the distances between all sample pairs are the same. Although minimizing the negative term gives uniformity, the positive term is also needed for aligning similar pairs, as we observe in §4. This implies the *tradeoff* between alignment and uniformity.

In fact, eq. (20) provides us guidance to select proper $f$-divergences for uniformity. In Table 1, we list some common choices of $f$-divergences. By inspecting the last column and using the definition of $G_\sigma$, we can easily verify that they all satisfy eq. (20). However, this is not true for all $f$-divergences. In Appendix A.1 we also provide some counterexamples that violate eq. (20) and thus Theorem 4, such as the Reversed Kullback–Leibler (RKL) and the Neynman $\chi^2$ divergences. Experimentally, we found that these divergences generally result in feature collapse (*i.e.*, all feature vectors are the same) and thus poor performance in downstream applications.

## 4 Experiments

In this section, we empirically evaluate the analysis on our provided answers: (1) Can we go beyond the KL-based objective (§4.2): we apply various $f$-MICL objectives to popular vision and language datasets. In particular, we show that under the same network architecture design, $f$-MICL can always provide a better choice of objective. We observe that the best-performing $f$-divergence is largely dataset dependent. (2) Can we design a better similarity function (§4.3): we show that the proposed $f$-Gaussian similarity is more powerful than the heuristic cosine similarity, regardless of the choice of $f$. Moreover, we confirm empirically that $f$-MICL extends the nice property of alignment and uniformity in §4.4.

### 4.1 Experimental settings

Our detailed settings can be found in Appendix D. In all our experiments, we change only the objective of different methods for fair comparison. We use the $f$-Gaussian similarity in $f$-MICL by default.

**Vision task.** Our vision datasets include CIFAR-10 (Krizhevsky et al., 2009), STL-10 (Coates et al., 2011), TinyImageNet (Chrabaszcz et al., 2017), and ImageNet (Deng et al., 2009) for image classification. After learning the feature embeddings, we evaluate the quality of representations using the test accuracy via a linear classifier. Note that we use $\alpha = 40$ across all vision experiments.
(1) Smaller datasets: For feature encoders, we use ResNet-18 (He et al., 2016) for CIFAR-10; ResNet-50 (He et al., 2016) for the rest. Our implementation is based on SimCLR (Chen et al., 2020), where we used the same 3-layer projection head during training. All models are trained for 800 epochs.
(2) ImageNet: We choose Vision Transformer (ViT-S) (Dosovitskiy et al., 2020) as our feature encoder. We choose the smaller ViT-S model with 6 blocks because larger ViT models are extremely expensive to train on GPUs. Our implementations are based on MoCo V3 (Chen et al., 2021), where models are trained for 1000 epochs.

Table 2: We compare the test accuracy (%) obtained with the linear evaluation on the vision datasets. On the Wikipedia dataset, we compare the semantic textual similarity (STS) via the Spearman's correlation. For each dataset and each method we take three different runs to get the mean and the standard derivation.

| Dataset | Baselines | | | $f$-MICL | | | |
|---|---|---|---|---|---|---|---|
| | MoCo | SimCLR | AU | KL | JS | Pearson | VLC |
| CIFAR-10 | $90.30_{\pm 0.19}$ | $89.71_{\pm 0.37}$ | $90.41_{\pm 0.26}$ | $\mathbf{90.61_{\pm 0.47}}$ | $89.66_{\pm 0.28}$ | $89.35_{\pm 0.52}$ | $89.13_{\pm 0.33}$ |
| STL-10 | $83.69_{\pm 0.22}$ | $82.97_{\pm 0.32}$ | $84.44_{\pm 0.19}$ | $85.33_{\pm 0.39}$ | $\mathbf{85.94_{\pm 0.17}}$ | $82.64_{\pm 0.37}$ | $\mathbf{85.94_{\pm 0.72}}$ |
| TinyImageNet | $35.72_{\pm 0.17}$ | $30.56_{\pm 0.28}$ | $41.20_{\pm 0.19}$ | $39.46_{\pm 0.20}$ | $42.98_{\pm 0.18}$ | $\mathbf{43.45_{\pm 0.54}}$ | $38.65_{\pm 0.45}$ |
| Wikipedia | $77.88_{\pm 0.15}$ | $77.40_{\pm 0.12}$ | $77.95_{\pm 0.08}$ | $\mathbf{78.02_{\pm 0.13}}$ | $76.76_{\pm 0.09}$ | $77.59_{\pm 0.12}$ | $55.07_{\pm 0.13}$ |

Table 3: We compare the test accuracy (%) with SOTA methods on ImageNet. We take three different runs to get the mean, where the standard derivations are less than 0.1% for $f$-MICL.

| Dataset | Baselines | | | | | | $f$-MICL | | |
|---|---|---|---|---|---|---|---|---|---|
| | SwAV | BYOL | Barlow Twins | VICReg | RényiCL | MoCo v3 | KL | JS | Pearson |
| ImageNet | 75.3 | 74.3 | 73.2 | 73.2 | **76.2** | 73.2 | 73.9 | **76.5** | 74.6 |

**Language task.** To show the wide applicability of our $f$-MICL framework, we also conduct experiments on a natural language dataset, English Wikipedia (Gao et al., 2021). We follow the experimental setting in (Gao et al., 2021), which applies BERT-based models to SimCLR (Devlin et al., 2019; Liu et al., 2019). Specifically, we choose the BERT$_{\mathsf{base}}$ model due to limited computing resources. For $f$-MICL objectives, we choose $\alpha = 409600$. The application task is called semantic textual similarity (STS (Agirre et al., 2013)) and we report the averaged Spearman's correlation in Table 2 for comparison.

### 4.2 $f$-MICL objectives

**Smaller datasets and language task.** We first compare $f$-MICL with several InfoNCE-based contrastive learning algorithms (i.e., SimCLR (Chen et al., 2020), MoCo (He et al., 2020), and AU (Wang & Isola, 2020)) on smaller datasets and the language task in Table 2. Here we choose four $f$-divergences with the best overall performance. See Appendix D for results on other $f$-divergences.

From Table 2 we observe that: (1) As we have shown in §3.4 that $f$-MICL generalizes InfoNCE-based objectives, empirically KL-MICL achieves similar performance to the baselines. In practice, we can tune the hyperparameter $\alpha$ such that KL-MICL outperforms the InfoNCE-based objectives. (2) KL-MICL is not always the optimal choice. We can see that the best-performing $f$-MICL objectives refer to four different $f$-divergences on four datasets.

The above results indicate that $f$-MICL can provide a wide range of objective choices for the downstream tasks. Although how to derive an optimal $f$-divergence deserves more study in theory, in practice we can select the best $f$ among several common $f$-divergences on a validation set.

Besides the $f$-divergences in Table 1, in Theorem 4 we have identified non-satisfying $f$-divergences. In our experiments, we found that applying these $f$-divergences such as the RKL and Neyman $\chi^2$ divergences would result in feature collapse. For example, in Figure 4 we show that with RKL the features all collapse to a constant.

**ImageNet results:** To further demonstrate the efficacy of $f$-MICL we then compare with several popular self-supervised learning methods, including both contrastive-based and non contrastive-based ones. These methods can be categorized into the ResNet-based (SwAV (Caron et al., 2020), BYOL (Huynh et al., 2020), Barlow Twins (Zbontar et al., 2021), VICReg (Bardes et al., 2021)) and RényiCL (Lee & Shin, 2022), and the ViT-based (Dosovitskiy et al., 2020). For the ResNet-based methods, we directly retrieve results from (Bardes et al., 2021), which are obtained by training ResNet-50 models for 1000 epochs. Chen et al. (Chen

Table 4: Comparison between the cosine and $f$-Gaussian similarities on CIFAR-10 with the test accuracy (%). For the Tsallis-$\alpha$ divergence we take $\alpha = 3$.

| Similarity | KL | JS | Pearson | SH | Tsallis-$\alpha$ | VLC |
|---|---|---|---|---|---|---|
| Cosine | 89.95 $\pm 0.26$ | 88.06 $\pm 0.33$ | 87.79 $\pm 0.42$ | 87.06 $\pm 0.55$ | 88.55 $\pm 0.28$ | 10.00 $\pm 0.00$ |
| Gaussian | **90.61** $\pm 0.47$ | **89.66** $\pm 0.28$ | **89.35** $\pm 0.52$ | **89.52** $\pm 0.25$ | **89.15** $\pm 0.42$ | **89.13** $\pm 0.33$ |

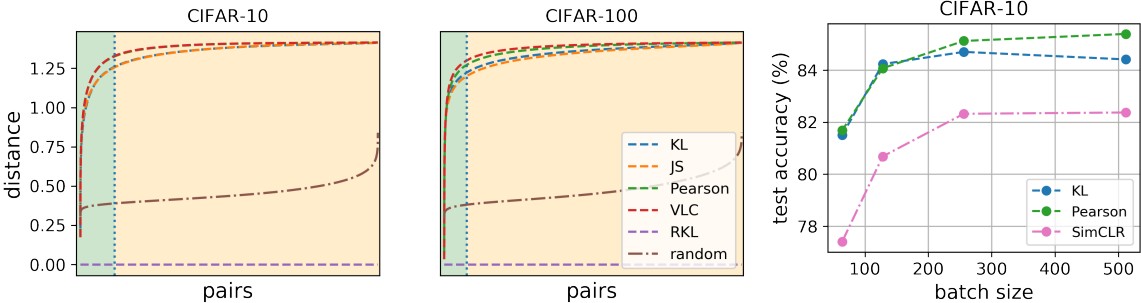

Figure 4: **(left and middle)** Distances between pairs of normalized features within a batch. **Green region:** similar pairs. **Orange region:** dissimilar pairs. $f$-MICL gives nearly uniform distances for dissimilar pairs for the $f$-divergences in Table 1. For non-satisfying $f$-divergences such as the RKL, the features collapse to a constant and thus the distances are zero. **(right)** The test accuracy v.s. the batch size after training 200 epochs for all algorithms.

et al., 2021) show that these two types of methods are directly comparable in terms of the model size and supervised learning performance. For the ViT-based method and our $f$-MICL, we apply ViT-S for 1000 epochs. Specifically, our $f$-MICL follows MoCo v3 with only the objective changed for different choices of the $f$-divergence.

Results in Table 3 show that: (1) By only changing the objective function, our method improves MoCo v3 by 2.1% using JS-MICL. (2) $f$-MICL objectives are comparable with the ResNet-based methods (e.g., SwAV and RényiCL).

Overall, our experiments confirm that $f$-MICL can provide a better choice of objective than InfoNCE on a variety of datasets, tasks, and encoder architectures.

### 4.3 $f$-Gaussian similarity

Next, we want to examine the effect of our similarity function while fixing the $f$-divergences. In Table 4 we compare the cosine and $f$-Gaussian similarities for different $f$-divergences on CIFAR-10. It can be seen that under our $f$-MICL framework the $f$-Gaussian similarity consistently outperforms the cosine similarity for various $f$-divergences[5]. This agrees with our Theorem 2 and eq. (9), and also implies the validity of Assumption 3. In particular, we identify that without the theoretical guarantee, the heuristic cosine similarity would fail for certain $f$-MICL objectives (*e.g.*, VLC).

### 4.4 Alignment and uniformity test

We empirically check the properties of alignment and uniformity for $f$-MICL by plotting the pairwise distance $\|x_i^g - x_j^g\|$ of the feature representations within the same batch on CIFAR-10. We compute the distances

---

[5]As we have shown the equivalence between cosine and Gaussian similarity on KL, the difference results on KL just show the choice of different scaling factors.

between the normalized features of every pair from a random batch, and then sort the pairs in increasing order. From Figure 4 we can see that $f$-MICL gives nearly uniform distances for dissimilar pairs (orange regions) on both datasets with various proper $f$-divergences. In contrast, a random initialized model gives a less uniform distribution for dissimilar pairs. Besides, we observe small pairwise distances for similar pairs (green regions).

## 4.5 Sensitivity to batch size

Finally, we study the sensitivity to the batch size of our $f$-MICL framework on CIFAR-10. On the right panel of Figure 4, we evaluate the classification accuracy by varying the batch size for different $f$-divergences and SimCLR. We can see that for all different batch sizes and with the proper choice of $f$-divergences, our performance is always better than SimCLR. In other words, we require fewer negative samples to achieve the same performance.

## 5   Related Work

**Contrastive learning.**   Self-supervised contrastive learning learns representations by contrasting sample pairs. Recently it has been shown analytically that improving the contrastiveness can benefit the downstream applications (Saunshi et al., 2019; Tosh et al., 2021). For popular contrastive learning methods such as Contrastive Predictive Coding (CPC) (van den Oord et al., 2018), SimCLR (Chen et al., 2020), and MoCo (He et al., 2020), their loss functions can be interpreted as a lower bound of mutual information, which is essentially the KL divergence between the joint distribution and the product of margin distributions. Besides the KL divergence, other statistical divergences or distances have been individually studied under the context of contrastive learning, *e.g.*, the Wasserstein distance (Ozair et al., 2019), Pearson $\chi^2$ divergence (Tsai et al., 2021), and Jensen–Shannon divergence (Hjelm et al., 2018).

**$f$-divergences** have been widely used in generative models (Nowozin et al., 2016) and domain adaptation (Acuna et al., 2021), for measuring the discrepancy of two distributions, where the variational lower bound is often employed for estimation. Compared to $f$-GAN (Nowozin et al., 2016) and $f$-DAL (Acuna et al., 2021) which minimize the $f$-divergence between two different distributions, our $f$-MICL objective is to maximize the $f$-divergence between the joint distribution and the product of marginal distributions. This agrees with our purpose of contrasting sample pairs. Moreover, we provide a theoretical criterion for choosing proper $f$-divergences.

**Mutual Information** also plays an important role in the context of deep representation learning (Tian et al., 2020a; Bachman et al., 2019; Hjelm et al., 2018; Tian et al., 2020b; Poole et al., 2019; Belghazi et al., 2018). *Loss function wise*, our losses partially cover the losses in the literature and generalizes them: e.g., Poole et al. (2019) considers several variational lower bounds of mutual information, where we generalize the DV objective; (b) *application-wise*, none of them considers contrastive learning: e.g., Poole et al. (2019) considers mutual information estimation, Belghazi et al. (2018) improves adversarial generative models, Hjelm et al. (2018) considers representation learning that maximizes local and global information.

**Metric learning.** Our work is closely related to metric learning (Kaya & Bilge, 2019; Suárez-Díaz et al., 2018), which aims to learn a distance metric bringing similar objects closer and distancing dissimilar objects further. In contrastive learning, a pre-defined similarity metric, *e.g.*, the cosine similarity (Chen et al., 2020; He et al., 2020) or a bilinear function (van den Oord et al., 2018; Tian et al., 2020a; Hénaff et al., 2020) is commonly used to measure the sample similarity. These pre-designed metrics may not necessarily lead to satisfactory performance in practice. Comparably, the design of our similarity function is empirically tailored for contrastive learning.

Finally, we summarize existing representation learning methods that utilize $f$-divergences and compare with $f$-MICL in Table 5 for a clear view of the literature.

Table 5: Comparison between different representation learning methods that apply $f$-divergences.

| Method | Objective | $f$-divergence | Similarity | Task |
|---|---|---|---|---|
| CPC (van den Oord et al., 2018) | InfoNCE | KL | log-bilinear | predictive coding |
| RPC (Tsai et al., 2021) | RPC | Pearson $\chi^2$ | cosine | predictive coding |
| MINE (Belghazi et al., 2018) | DV bound of MI | KL | neural net | GAN generation |
| DIM (Hjelm et al., 2018) | JSD/InfoNCE | JS & KL | neural net | representation learning |
| Poole et al. (2019) | MI bounds | KL | joint/separable | representation learning |
| SimCLR (Chen et al., 2020) | InfoNCE | KL | cosine | contrastive learning |
| MoCo (He et al., 2020) | InfoNCE | KL | cosine | contrastive learning |
| AU (Wang & Isola, 2020) | InfoNCE | KL | Gaussian | contrastive learning |
| **$f$-MICL (Ours)** | $f$-MI | general | $f$-Gaussian | contrastive learning |

## 6 Conclusion

We developed $f$-MICL for contrastive learning, which generalizes the KL-based mutual information to the $f$-mutual information. With $f$-MICL we provided a broad spectrum of objective choices with better downstream performance. We also proposed a novel $f$-Gaussian similarity function, which shows superior performance to the commonly used cosine similarity. In addition, we confirmed the generalization of $f$-MICL by comparing with popular InfoNCE-based objectives. Empirically, we exhibited the efficacy of $f$-MICL across a wide range of datasets from both vision and natural language.

**Limitations and future work.** While $f$-MICL provides a variety of objective functions, it is yet unclear how to choose an optimal $f$ based on a task and a dataset in theory, such that we usually rely on a validation set in practice for selection. An interesting future work is to learn an optimal $f$-divergence using a parametrized neural network. Moreover, Lee & Shin (2022) applied Skew Rényi divergence for contrastive learning. However, we observe that applying Rényi-MICL naively leads to a large variance (similar to Section 4.1 in Lee & Shin 2022), and we leave the discussion on skew divergences for future works. Additionally, McAllester & Stratos (2020) showed that there exist some inherent statistical limitations on accurately estimating the mutual information with various lower bounds. In future work it would be interesting to examine if such limitations extend to $f$-MI, and if a limited estimation of $f$-MI necessarily affects $f$-MICL whose goal is to compare and learn representations through (lower bounds of) $f$-MI.

## Acknowledgement

We thank the reviewers and the action editor for their constructive comments. Part of this work was performed during YL's internship at NRC. YY thanks NSERC and CIFAR for funding support. Resources used in preparing this research were provided, in part, by the Province of Ontario, the Government of Canada through CIFAR, and companies sponsoring the Vector Institute.

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

Table 6: A summary of common $f$-divergences. KL: Kullback–Leibler; JS: Jensen–Shannon; SH: Squared Hellinger. For JS, we define $\varphi(u) = -(u+1)\log\frac{1+u}{2} + u\log u$. For Pearson $\chi^2$, we take $f^*(t) = -1$ if $t \le -2$. For Jeffrey, $\widehat{W} = W + W^{-1}$ and $W(\cdot)$ is the Lambert-$W$ product log function. The Tsallis-$\alpha$ divergence is defined in Tsallis (1988) and we have $\alpha > 1$ for $f$-divergences. We ignore constant addition $-1/(\alpha - 1)$ because it does not change the optimization problem. The Vincze–Le Cam divergence can be found in Le Cam (2012) which is closely related to $\chi^2$ and Hellinger divergences. For the Vincze–Le Cam divergence we require $-3 < t < 1$ and $f^*(t) = -1$ if $t \le -3$.

| **Divergence** | $f(u)$ | $f^*(t)$ | $f'(u)$ | $f^* \circ f'(u)$ |
|---|---|---|---|---|
| KL | $u\log u$ | $\exp(t-1)$ | $\log u + 1$ | $u$ |
| Reverse KL | $-\log u$ | $-1 - \log(-t)$ | $-1/u$ | $\log u - 1$ |
| JS | $\varphi(u)$ | $-\log(2 - e^t)$ | $\log 2 + \log\frac{u}{1+u}$ | $-\log 2 + \log(1+u)$ |
| Pearson $\chi^2$ | $(u-1)^2$ | $t^2/4 + t$ | $2(u-1)$ | $u^2 - 1$ |
| SH | $(\sqrt{u} - 1)^2$ | $\frac{t}{1-t}$ | $1 - u^{-1/2}$ | $u^{1/2} - 1$ |
| Neyman $\chi^2$ | $\frac{(1-u)^2}{u}$ | $2 - 2\sqrt{1-t}$ | $1 - u^{-2}$ | $2 - 2u^{-1}$ |
| Jeffrey | $(u-1)\log u$ | $\widehat{W}(e^{1-t}) + t - 2$ | $1 - u^{-1} + \log u$ | $\widehat{W}(e^{1/u}/u) + \log u - \frac{1+u}{u}$ |
| Tsallis-$\alpha$ | $\frac{u^\alpha}{\alpha-1}$ | $(\frac{\alpha-1}{\alpha}t)^{\frac{\alpha}{\alpha-1}}$ | $\frac{\alpha}{\alpha-1}u^{\alpha-1}$ | $u^\alpha$ |
| Vincze–Le Cam | $\frac{(u-1)^2}{u+1}$ | $4 - t - 4\sqrt{1-t}$ | $\frac{(u-1)(u+3)}{(u+1)^2}$ | $3 - \frac{4}{u+1}$ |

# A   Additional theoretical results

In this appendix, we provide additional theoretical results, including additional $f$-divergences and the theory for weighting parameters.

**Notations.** We assume that a dominating measure $\lambda$ (e.g. Lebesgue) is given and all other probability measures are represented as some density w.r.t. $\lambda$. Given the joint density $p(x,y)$, we denote $p(x) = \int p(x,y)\mathrm{d}\lambda(y)$ and $p(y) = \int p(x,y)\mathrm{d}\lambda(x)$ as the marginals. We use $\mathrm{supp}(\cdot)$ to denote the support of a distribution, and $f^*$ to denote the conjugate of function $f$. Every norm presented is Euclidean. We use $x^g := g(x)$ as the shorthand notation for the feature embedding, with $x$ a raw sample. The notation $p_d$ stands for the data distribution, and $p_\times := p_d \otimes p_d$ means its self product. We denote $p_+$ as the distribution of *positive pairs*, *i.e.*, two samples with similar feature embeddings. The symbol $\circ$ denotes function composition.

## A.1   Additional $f$-divergences

We expand Table 1 and give more examples of $f$-divergences in Table 6. As we will see in the proof of Theorem 4, Table 1 gives a special class of $f$-divergences that guarantees uniformity. A detailed description of $f$-divergences can be found in e.g. Sason & Verdú (2016).

## A.2   Weighting parameters

In Algorithm 1 we added a weighting parameter $\alpha$ to balance the alignment and uniformity. We prove that even after adding this parameter we are still maximizing the $f$-mutual information, although with respect to a different $f$.

**Proposition 5 (weighting parameter).** *Given $\alpha > 0$ and a closed convex function $f : \mathbb{R}_+ \to \mathbb{R}$ such that $f(1) = 0$, define $f_\alpha : \alpha\, \mathrm{dom}\, f \to \mathbb{R}$ with*

$$f_\alpha(x) = \alpha f\left(\frac{x}{\alpha}\right) - \alpha f\left(\frac{1}{\alpha}\right)$$

*for any $x \in dom\ f$. Then $I_{f_\alpha}$ is still a valid $f$-mutual information (see Definition 1). Besides, by replacing $f$ with $f_\alpha$ in eq. (10) we have the following optimization problem:*

$$\sup_{g \in \mathcal{G}} \mathbb{E}_{(x,y) \sim p_+} \left[ f' \left( \frac{G_\sigma(\|x^g - y^g\|^2)}{\alpha} \right) \right] - \alpha \mathbb{E}_{(x,y) \sim p_\times} \left[ f^* \circ f' \left( \frac{G_\sigma(\|x^g - y^g\|^2)}{\alpha} \right) \right],$$

*where $G_\sigma(\|x^g - y^g\|^2) = \mu \exp \left( -\frac{\|x^g - y^g\|^2}{2\sigma^2} \right)$ is the Gaussian kernel.*

Note that $\alpha\, dom\ f$ means the scalar multiplication of a set which is applied element-wisely. According to Definition 1, $f_\alpha$ is also a valid $f$-divergence. This proposition tells us that rescaling the second term with factor $\alpha$ is equivalent to changing the function $f$ to another convex function $f_\alpha$. The transformation from $f$ to $\alpha f\left(\frac{x}{\alpha}\right)$ is also known as right scalar multiplication (Urruty & Lemaréchal, 1993). Let us now move on to our proof:

*Proof.* By definition, we know that $f_\alpha$ is convex and closed with $f_\alpha(1) = 0$, and thus $I_{f_\alpha}$ is a valid $f$-mutual information according to Definition 1. Moreover, we have $f'_\alpha(x) = f'(\frac{x}{\alpha})$ for any $x \in \alpha\, dom\ f$ and

$$
\begin{aligned}
f_\alpha^*(t) &= \sup_{x \in dom\ f_\alpha} xt - f_\alpha(x) \\
&= \sup_{x \in \alpha dom\ f} xt - \alpha f\left(\frac{x}{\alpha}\right) + \alpha f\left(\frac{1}{\alpha}\right) \\
&= \sup_{\frac{x}{\alpha} \in dom\ f} \frac{x}{\alpha} \cdot (\alpha t) - \alpha f\left(\frac{x}{\alpha}\right) + \alpha f\left(\frac{1}{\alpha}\right) \\
&= \alpha \sup_{\frac{x}{\alpha} \in dom\ f} \left( \frac{x}{\alpha} \cdot t - f\left(\frac{x}{\alpha}\right) \right) + \alpha f\left(\frac{1}{\alpha}\right) \\
&= \alpha f^*(t) + \alpha f\left(\frac{1}{\alpha}\right),
\end{aligned}
\tag{21}
$$

where in the last line we used the definition of $f^*(t)$. Plugging $f'_\alpha$ and $f_\alpha^*$ into eq. (10) yields the desired result. $\qquad\square$

## B  Proofs

**Lemma 2** (*e.g.*, Nguyen et al. 2010, Lemma 1). *Suppose $f$ is differentiable, and the embedding function $g$ is fixed. The following similarity function $s_\star$ maximizes eq. (4):*

$$s_\star(x^g, y^g) = f' \left( \frac{p_g(x^g, y^g)}{p_g(x^g)p_g(y^g)} \right). \tag{5}$$

*Proof.* From Definition 1, we are computing the following supremum:

$$\sup_{g,s} \int \left( \frac{p_g(x^g, y^g)}{p_g(x^g)p_g(y^g)} s(x^g, y^g) - f^* \circ s(x^g, y^g) \right) dp_d^g \otimes p_d^g. \tag{22}$$

Suppose $s$ is unconstrained and we fix $g$. The optimal solution should satisfy:

$$\frac{p_g(x^g, y^g)}{p_g(x^g)p_g(y^g)} \in (\partial f^*)(s_\star(x^g, y^g)), \tag{23}$$

almost surely for $(x,y) \sim p_d \otimes p_d$. From (3.11) of Rockafellar (1966) this is equivalent to:

$$s_\star(x^g, y^g) \in \partial f \left( \frac{p_g(x^g, y^g)}{p_g(x^g)p_g(y^g)} \right). \tag{24}$$

If $f$ is differentiable, then for any $u \in dom\ f$, $\partial f(u) = \{f'(u)\}$ is a singleton. $\qquad\square$

**Theorem 4** (**Uniformity**). *Suppose that the batch size $N$ satisfies $2 \leq N \leq d+1$, with $d$ the dimension of the feature space. If the real function*

$$h(t) = f^* \circ f' \circ G_\sigma(t) \text{ is strictly convex on } [0, 4], \tag{20}$$

*then all minimizers of the second term of eq. (11), i.e., $\sum_{i \neq j} f^* \circ s_f(x_i^g, x_j^g)$, satisfy the following condition: the feature representations of all samples are distributed uniformly on the unit hypersphere $\mathbb{S}^{d-1}$.*

Note that we say the samples are "distributed uniformly" if the feature vectors form a regular simplex (see Figure 5), and thus the distances between all sample pairs are the same.

*Proof.* From the definition of $h$ it is clear that $h$ is decreasing since $f^*$ and $f'$ are both monotonically increasing white $G_\sigma$ is decreasing. Using $h$ we rewrite the second term of eq. (11) as

$$\min_{x_1^g, \dots, x_N^g \in \mathbb{S}^{d-1}} \sum_{i,j} h(\|x_i^g - x_j^g\|^2). \tag{25}$$

When $N \in [2, d+1]$, there exists a neat characterization of the minimizers, see e.g. Borodachov et al. (2019). We include the proof below for completeness.

Apply Jensen's inequality, we have:

$$
\begin{aligned}
\frac{1}{N^2} \sum_{i,j} h(\|x_i - x_j\|^2) &\geq h\left(\frac{1}{N^2} \sum_{i,j} \|x_i - x_j\|^2\right) \\
&= h\left(\frac{1}{N^2} \sum_{i,j} \|x_i - x_j\|^2\right) \\
&= h\left(\frac{1}{N^2} \sum_{i,j} (2 - 2x_i \cdot x_j)\right) \\
&= h\left(2\left(1 - \left\|\frac{1}{N} \sum_{i=1}^{N} x_i\right\|^2\right)\right) \\
&\geq h(2), \tag{26}
\end{aligned}
$$

where in the first line we used Jensen's inequality; in the third line we used $\|x_i\| = \|x_j\| = 1$ for any $i, j \in [N]$; in the last line we note that $\|\sum_{i=1}^{N} x_i\| \geq 0$ and $h$ is a decreasing function. When $h$ is strictly convex and decreasing, it is in fact strictly decreasing, and hence the two inequalities above can be attained iff

$$\bar{x} := \frac{1}{N} \sum_i x_i = \mathbf{0}, \quad \text{and } \|x_i - x_j\|^2 \equiv c \text{ for all } i \neq j, \tag{27}$$

namely that $\{x_1, \dots, x_N\}$ form a regular simplex with its center at the origin. We remark that when $h$ is merely convex, points forming a centered regular simplex may form a strict subset of the minimizers.

To see the necessity of $N \leq d+1$, let us note that

$$x_i^\top x_j = \begin{cases} 1, & i = j \\ -\frac{1}{N-1}, & i \neq j \end{cases}, \tag{28}$$

since

$$
\begin{aligned}
\sum_{ij} \|x_i - x_j\|^2 = 2N^2 = N(N-1)c &\implies \\
c = \frac{2N}{N-1} = 2 + \frac{2}{N-1}. &
\end{aligned} \tag{29}
$$

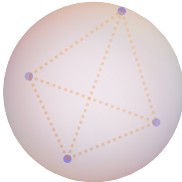

Figure 5: A regular simplex on a hypersphere.

Performing simple Gaussian elimination we note that the matrix $X^\top X$ has rank $N - 1$ where $X = [x_1, \ldots, x_N] \in \mathbb{R}^{d \times N}$. Therefore, we must have $N - 1 \leq d$.

Lastly, we need to show when $h$ is a (strictly) convex function, which may not always be true depending on the $f$-divergences. We give the following characterization (we ignore the constants $\mu$ and $2\sigma^2$ in Assumption 3 as they do not affect convexity):

- $h$ strictly convex: $h_{\mathrm{KL}}(t) = e^{-t}$, $h_{\mathrm{JS}}(t) = \log(1+e^{-t}) - \log 2$, $h_{\mathrm{Pearson}}(t) = e^{-2t} - 1$, $h_{\mathrm{SH}}(t) = e^{-t/2} - 1$, $h_{\mathrm{Tsallis}}(t) = e^{-\alpha t}$, $h_{\mathrm{VLC}} = 3 - \frac{4}{1+e^{-t}}$;
- $h$ convex but not strictly convex: $h_{\mathrm{RKL}}(t) = -t - 1$ (RKL stands for Reversed Kullback–Leibler, see Appendix A.1);
- $h$ concave: $h_{\mathrm{Neyman}}(t) = 2 - 2e^t$ (Neyman stands for Neyman $\chi^2$, see Appendix A.1).

Only for the last case we do not have the guarantee that the minimizing configurations could form a regular simplex. For RKL, in fact, any configuration that centers at the origin suffices since $h$ is a linear function. $\qquad\square$

## C  Estimation of $f$-MICL Objective

In § 3.3 we have provided the empirical estimation of our objective in eq. (11). However, it remains a question whether our estimation of $f$-mutual information is consistent. In this part we derive an upper bound for the estimation error from statistical learning theory.

We denote our $f$-MICL objective as $i_f(X; Y)$ (eq. (10)) and its empirical estimation as $\widehat{i}_f(X; Y)$ (eq. (11)). After fixing the similarity function , we have $T(x, y) = s_f(x^g, y^g) = f' \circ G_\sigma(\|x^g - y^g\|^2)$.

**Theorem 6** (estimation error). *Suppose that the function $T$ is taken from a function class $\mathcal{T}$ and define $\mathcal{T}_x$ as the function class of $T(x, \cdot)$ given some $x \in supp(p_d)$. Denote $\mathfrak{R}_N^P$ to be the Rademacher complexity w.r.t. the distribution $P$ with $N$ i.i.d. drawn samples. Then for any $T \in \mathcal{T}$, the estimation error $|i_f(X; Y) - \widehat{i}_f(X; Y)|$ is upper bounded with probability at least $1 - \delta$:*

$$2\mathfrak{R}_N^{p_+}(\mathcal{T}) + 2\mu \left( \mathop{\mathbb{E}}_{x \sim p_d} \mathfrak{R}_N^{p_d}(\mathcal{T}_x) + \frac{1}{N} \sum_{i=1}^{N} \mathfrak{R}_{N-1}^{p_d}(\mathcal{T}_{x_j}) \right) + (r_T + 2r_f) \sqrt{\frac{\log 6/\delta}{2(N-1)}}, \tag{30}$$

*with the constants $r_T = f'(\mu) - f'(\mu e^{-2/\sigma^2})$ and*

$$r_f = f^* \circ f'(\mu) - f^* \circ f'(\mu e^{-2/\sigma^2}).$$

Here the constant $\mu$ is from our Gaussian kernel in Assumption 3. Rademacher complexity evaluates the richness of a class of real-valued functions regarding a probability distribution, and its formal definition can be found in Koltchinskii (2001).

**Non-i.i.d. proof.** Our conclusion is theoretically non-trivial since our sample pairs are *non-i.i.d.*: although the individual samples are assumed to be i.i.d., the negative pairs are not independently drawn (*e.g.*, $(x_1, x_2)$ and $(x_1, x_3)$), which makes the derivation challenging.

Note that the function class $\mathcal{T}$ depends on the class of the feature encoder $g$ and the $f$-divergence. Our estimation error eq. (30) is composed of three parts:

- the Rademacher complexity of the function class $\mathcal{T}$. In general, if $\mathcal{T}$ is richer then its Rademacher complexity is also larger.
- the expected Rademacher complexity of the one-side function class $\mathcal{T}_x$ and its empirical estimation;
- an error term that decreases with more samples.

Since the encoders are usually built with neural networks, we can use the existing theory (Bartlett et al., 2019) to give more detailed bounds for the Rademacher complexities of $\mathcal{T}$. Specifically, if the Vapnik–Chervonenkis (VC) dimension of $\mathcal{T}$ is finite, then our estimation error in eq. (30) goes to zero as $N \to \infty$ (Mohri et al., 2018).

**Approximation and estimation tradeoff.** In order to minimize the estimation error in eq. (30), we should choose a simpler function class $\mathcal{T}$ to reduce the Rademacher complexities. However, $\mathcal{T}$ should also be rich enough so that eq. (3) can be satisfied, since our objective $i_f(X;Y)$ should approximate the $f$-mutual information $I_f(X;Y)$ if we choose the optimal $T$. Therefore, there is a natural tradeoff between approximation and estimation errors when we change the complexity of $\mathcal{T}$.

# D   Additional experimental results

We present additional experiment details in this appendix, to further support our experiments in the main paper.

## D.1   Implementation details

In this paper, we follow the implementations in SimCLR (`https://github.com/sthalles/SimCLR`) and MoCo v3 (`https://github.com/facebookresearch/moco-v3`). For vision tasks, we use ResNet (He et al., 2016) and ViT-S (Dosovitskiy et al., 2020) as the feature encoder, and we adopt the similar procedure of SimCLR/MoCo for sampling. For the language dataset, we follow the exact experimental setting of Gao et al. (2021) and only change the objective. Our experimental settings are detailed below:

- Hardware and package: We train on a GPU cluster with `NVIDIA T4` and `P100`. The platform we use is `pytorch`. Specifically, the pairwise summation can be easily implemented using `torch.nn.functional.pdist` from `pytorch`.
- Datasets: the datasets we consider include CIFAR-10, STL-10 (Coates et al., 2011), TinyImageNet (Chrabaszcz et al., 2017), ImageNet (Deng et al., 2009) and English Wikipedia (Gao et al., 2021).
- Augmentation method: For each sample in a dataset we create a sample pair, a.k.a. positive pair, using two different augmentation functions. For image samples, we choose the augmentation functions to be the standard ones in contrastive learning, e.g., in Chen et al. (2020) and He et al. (2020). The augmentation is a composition of random flipping, cropping, color jittering and gray scaling. For text samples, following the augmentation method of Gao et al. (2021) we use dropout masks.
- Neural architecture: For CIFAR-10 we use ResNet-18 (He et al., 2016); for STL-10, TinyImageNet we use ResNet-50 (He et al., 2016); for ImageNet we use ViT-S (Dosovitskiy et al., 2020); for the Wikipedia dataset we use BERT$_{\mathsf{base}}$ (Devlin et al., 2019).
- Batch size and embedding dimension: for experiments in CIFAR-10 we choose batch size 512; for STL-10 we choose batch size 64 to accommodate one GPU training; for TinyImageNet, we choose batch size 256; for ImageNet, we choose batch size 1024. For all the vision datasets, we choose the embedding dimension to be 512. Regarding the language dataset, the batch size is 64 with the feature dimension 768. In all of these cases, our assumption $N \le d + 1$ in Theorem 4 is satisfied.
- Hyperparameters: in all our experiments we fix the constant factor $\mu = 1$. We find that in practice the weight parameter $\alpha$ often needs to be large (*e.g.*, in the Wikipedia dataset), which requires moderate tuning.
- Optimizer and learning rate scheduler: For smaller vision tasks, we use SGD with momentum for optimization and the cosine learning rate scheduler (Loshchilov & Hutter, 2017). For the ImageNet

Table 7: Detailed experimental settings. `arch`: the neural network architecture used. $N$: batch size; $d$: the dimension of the feature representation; `lr`: learning rate; $\mu$: the constant factor in $\mu$; $1/(2\sigma^2)$ and $\alpha$ follow from Algorithm 1; `epoch`: the number of epochs we run; $k$: the number of nearest neighbors in $k$-NN evaluation.

| Dataset | `arch` | $N$ | $d$ | `lr` | $\mu$ | $(2\sigma^2)^{-1}$ | $\alpha$ | `epoch` | $k$ |
|---|---|---|---|---|---|---|---|---|---|
| CIFAR-10 | ResNet-18 | 512 | 512 | 0.1 | 1 | 1 | 40 | 800 | 200 |
| STL-10 | ResNet-50 | 64 | 512 | 0.1 | 1 | 1 | 40 | 800 | 200 |
| TinyImageNet | ResNet-50 | 256 | 512 | 0.1 | 1 | 1 | 40 | 800 | 200 |
| ImageNet | ViT-S | 1024 | 512 | 0.1 | 1 | 1 | 40 | 1000 | n/a |
| Wikipedia | BERT$_{\text{base}}$ | 64 | 768 | 3e-5 | 1 | 20 | 409600 | 1 | n/a |

task and natural language task, we use Adam with weight decay (Loshchilov & Hutter, 2018) and the linear decay scheduler.

- Evaluation metric: for vision tasks, we use $k$-nearest-neighbor ($k$-NN) (only small datasets) and linear evaluation to evaluate the performance, based on the learned embeddings. For the NLP task, we use the Spearman's correlation to evaluate the averaged semantic textual similarity score (Gao et al., 2021).
- Baseline methods: for the four baseline methods, we follow the implementations in:
  - MoCo: `https://github.com/facebookresearch/moco`;
  - SimCLR: `https://github.com/sthalles/SimCLR`;
  - Uniformity: `https://github.com/SsnL/align_uniform`;
  - MoCo v3: `https://github.com/facebookresearch/moco-v3`

  For fair comparison we use the experimental settings in Table 7 for all the baseline methods, which might differ from the original settings.

Table 7 gives common choices of hyperparameters for different datasets. Note that we may need to further finetune $\alpha$ and $\sigma$ for different $f$-divergences. See our supplementary code for more details.

### D.2 Additional ablation study on weighting parameter

We provide additional ablation study on the weighting parameter $\alpha$. We perform experiments using a vision dataset (CIFAR-10) and a language dataset (Wikipedia). For CIFAR-10, we vary $\alpha$ from 0.1 to 50 for KL and JS divergences and run for 200 epochs. We perform the same experiments on KL when $\alpha = 1$ for 800 epochs and observed an accuracy of 83.58% (lower than SimCLR). This observation further provides empirical evidence that KL-MICL is different from InfoNCE and needs special tuning on $\alpha$ to perform well.

Table 8 justifies our choice of $\alpha$ in Table 7, where the downstream test accuracy indicates the optimal performance when choosing $\alpha = 40$. For the Wikipedia dataset, we observe that a much bigger $\alpha$ is desirable for maximum performance. We vary $\alpha$ from 1 to $10^6$ for KL and Pearson $\chi^2$ divergences and run for 1 epoch, as there is a large number of samples ($10^6$) in the language dataset. Table 9 justifies our choice of $\alpha$ in Table 7, where the best performance is reached at $\alpha = 409600$. Such an $\alpha$ is found by starting from $\alpha = 100$ and doubling iteratively.

Table 8: Ablation study on weighting parameter $\alpha$ for KL and JS divergences on CIFAR-10. We compare test accuracies (%) for different choices of $\alpha$ using $k$-NN evaluation.

| $\alpha$ | 0.1 | 1 | 10 | 20 | 30 | 40 | 50 |
|---|---|---|---|---|---|---|---|
| KL | 13.16 | 77.60 | 83.53 | 83.77 | 81.39 | **84.19** | 82.77 |
| JS | 8.84 | 73.31 | 81.39 | 83.21 | 83.49 | **84.06** | 82.61 |

Table 9: Ablation study on weighting parameter $\alpha$ for KL and Pearson $\chi^2$ divergences on Wikipedia. We compare the semantic textual similarity (STS) via the Spearman's correlation for different choices of weighting parameter $\alpha$.

| $\alpha$ | 1 | 10 | $10^2$ | $10^3$ | $10^4$ | $10^5$ | 409600 | $10^6$ |
|---|---|---|---|---|---|---|---|---|
| KL | 67.52 | 70.47 | 72.43 | 75.12 | 76.90 | 77.78 | **78.02** | 77.78 |
| Pearson | 64.58 | 67.78 | 71.58 | 74.03 | 74.95 | 74.40 | **77.59** | 76.47 |

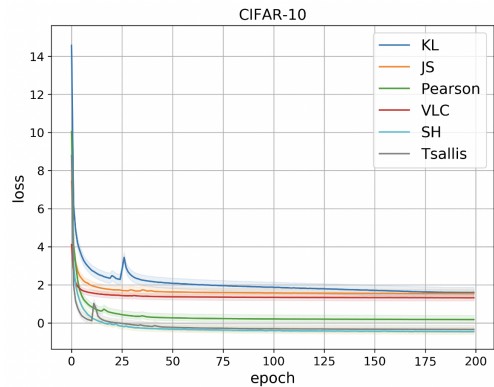

Figure 6: The training loss curves of various $f$-divergences on CIFAR-10 with 200 epochs.

## D.3 Additional experiments

Our final experiments show that $f$-MICL is stable in terms of training and the variation of performance is well controlled.

**Training stability** We depict the training loss curves of different divergences on CIFAR-10 in Figure 6. This figure shows that our methods exhibit stable training dynamics with fast convergence.

**$k$-NN evaluation and additional $f$-divergences** We show more detailed results of Table 2 in Table 10, including experiments using $k$-nearest neighbour ($k$-NN) evaluation. Additionally, we have added experiments on other $f$-divergences such as Squared Hellinger and Tsallis-$\alpha$ divergences.

**Verification of Assumption 3** Throughout our paper we made an assumption (Assumption 3) that the joint feature distribution is a Gaussian kernel. However, is it a valid assumption? In this experiment, we try to show some empirical evidence that this assumption approximately holds in practice. Recall that Assumption 3 says that the joint feature distribution of positive pairs is:

$$p_g(x^g, y^g) \propto \exp\left(-\frac{\|x^g - y^g\|^2}{2\sigma^2}\right) \tag{31}$$

if the RBF kernel is Gaussian. In order to estimate the joint density of positive pairs, we use normalizing flows, which is a popular method for density estimation. Popular normalizing flow models include NICE (Dinh et al., 2014), RealNVP (Dinh et al., 2017) and Glow (Kingma & Dhariwal, 2018). Equation (31) is equivalent to the following:

$$\log p_g(x^g, y^g) = -\frac{\|x^g - y^g\|^2}{2\sigma^2} + \text{const}, \tag{32}$$

and thus it suffices to show that the log likelihood is linear w.r.t. the distances between each positive pair. In Figure 7, we plot the relation between $\log p_g$, estimated by RealNVP [6] with a Gaussian prior, and the squared

---
[6]Code available at `https://github.com/ikostrikov/pytorch-flows`.

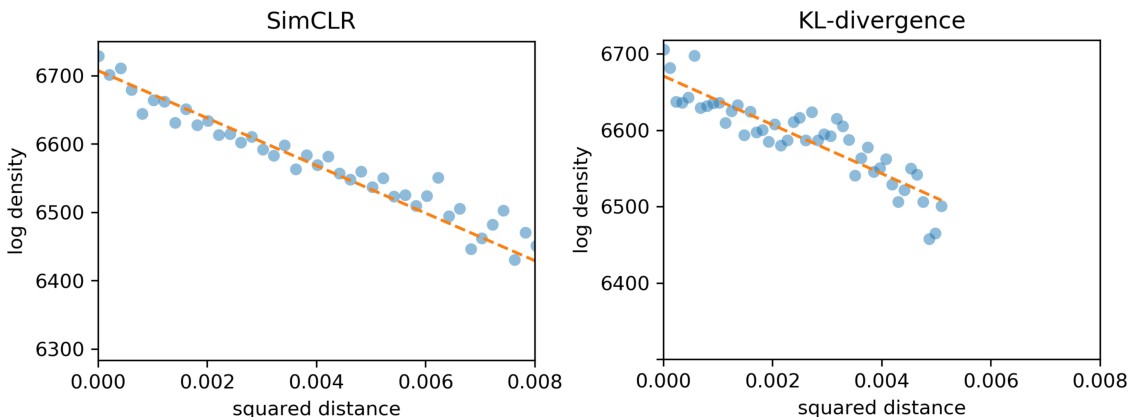

Figure 7: Experiment for verifying Assumption 3. We draw the relation between the squared distances $\|x^g - y^g\|^2$ and the averaged $\log p_g$ with RealNVP. The features are learned by different algorithms trained on CIFAR-10. (**left**) SimCLR; (**right**) $f$-MICL with the KL divergence.

Table 10: Test accuracy (%) on the smaller vision datasets. For the Wikipedia dataset we evaluate the semantic textual similarity (STS) via the Spearman's correlation. For each method, we take three separate runs, and show the mean and stand derivation.

| Evaluation | Dataset | Baselines | | | $f$-**MICL** | | | | | |
|---|---|---|---|---|---|---|---|---|---|---|
| | | MoCo | SimCLR | Uniformity | KL | JS | Pearson | SH | Tsallis | VLC |
| Linear | CIFAR-10 | 90.30 ±0.19 | 89.71 ±0.37 | 90.41 ±0.26 | **90.61** ±**0.47** | 89.66 ±0.28 | 89.35 ±0.52 | 89.52 ±0.25 | 89.15 ±0.42 | 89.13 ±0.33 |
| | STL-10 | 83.69 ±0.22 | 82.97 ±0.32 | 84.44 ±0.19 | 85.33 ±0.39 | **85.94** ±**0.17** | 82.64 ±0.37 | 82.80 ±0.27 | 84.79 ±0.34 | **85.94** ±**0.72** |
| | TinyImageNet | 35.72 ±0.17 | 30.56 ±0.28 | 41.20 ±0.19 | 34.95 ±0.20 | 42.98 ±0.18 | **43.45** ±**0.54** | 40.83 ±0.67 | 32.99 ±0.49 | 38.65 ±0.45 |
| $k$-NN | CIFAR-10 | 88.70 ±0.22 | 84.92 ±0.39 | 89.42 ±0.18 | 89.34 ±0.57 | 89.12 ±0.38 | **89.44** ±**0.60** | 88.13 ±0.18 | 89.18 ±0.62 | 89.15 ±0.23 |
| | STL-10 | 78.77 ±0.25 | 74.34 ±0.14 | 79.57 ±0.52 | 79.99 ±0.47 | **80.45** ±**0.19** | 76.64 ±0.26 | 78.31 ±0.33 | 76.11 ±0.24 | 79.34 ±0.62 |
| | TinyImageNet | 36.22 ±0.20 | 29.60 ±0.39 | 37.44 ±0.27 | 36.17 ±0.29 | **38.20** ±**0.26** | 38.14 ±0.63 | 35.56 ±0.77 | 33.11 ±0.52 | 35.21 ±0.33 |
| STS | Wikipedia | 77.88 ±0.15 | 77.40 ±0.12 | 77.95 ±0.08 | **78.02** ±**0.13** | 76.76 ±0.09 | 77.59 ±0.12 | 73.60 ±0.10 | 72.68 ±0.09 | 55.07 ±0.13 |

distances $\|x^g - y^g\|^2$. The representations are learned by SimCLR, and $f$-MICL with the KL divergence on the CIFAR-10 dataset. To alleviate the estimation error in the flow model, we divide the distances into small intervals and compute the average log-likelihood within each interval. We can see that the log-likelihood is roughly linear w.r.t. the squared distance, and thus verifying our Assumption 3. Moreover, in Figure 8, we also show the estimation by training RealNVP with a uniform prior. Over 5 different combinations of random data augmentations, we observe that the linear relationship generally holds and the estimations by Gaussian prior and uniform prior are very similar.

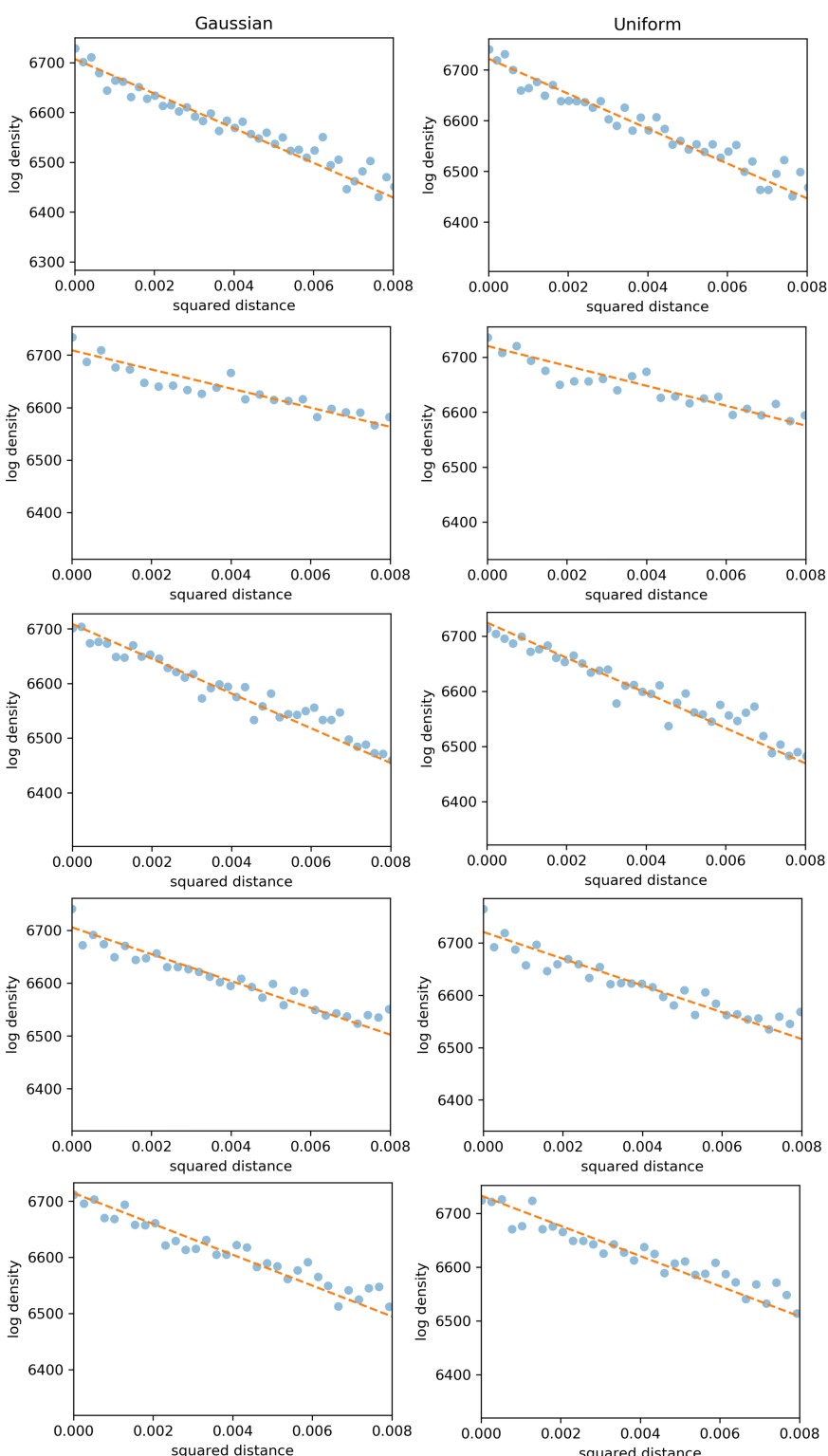

Figure 8: Additional experiment for verifying Assumption 3. Here we take 5 different combinations of random data augmentations and draw the relation between the squared distances $\|x^g - y^g\|^2$ and the averaged $\log p_g$ with RealNVP. (**left column**) Gaussian prior; (**right column**) Uniform prior. The features are learned by SimCLR trained on CIFAR-10.

