# OpenReview forum: "$f$-MICL: Understanding and Generalizing InfoNCE-based Contrastive Learning"
_TMLR — Accepted by TMLR_

### Review · Reviewer_DHPy · 2023-07-31

**Summary Of Contributions:**

In their paper "f-MICL: Understanding and Generalizing InfoNCE-based Contrastive Learning" the authors suggest a generalization of the InfoNCE loss function commonly used in self-supervised learning (SSL) methods such as SimCLR. They show that they can set up meaningful loss functions using many different f-divergences, obtaining a whole family of SSL losses. Empirically, they find that different losses perform differently well on different datasets (even though the differences are mostly moderate).

**Audience:**

Yes

**Broader Impact Concerns:**

No concerns.

**Claims And Evidence:**

Yes

**Requested Changes:**

MAJOR ISSUES:

* I was confused by the relationship between p+ (distribution of positive pairs) and p(x,y) (joint distribution). Are they the same thing? For example, in Assumption 3 you say that it's reasonable to assume that p(x,y) decreases with distance between x and y. This only makes sense if x and y are a positive pair. But you don't explicitly say that, and in general you use p+ notation for the distribution of positive pairs. Can you elaborate on the relationship between p+ and p(x,y) somewhere early on in Section 3?

* In Section 4, the quality of representation is assessed in which space? In SimCLR there is usually a projection head after the ResNet; the loss operates on the output of the projector, but the evaluation (linear classification accuracy) is assessed on the output of the ResNet before the projector. From your description it is unclear how you did it. Please clarify.

* Section 4.1 says "All models are trained for 200 epochs". However, Table 6 in the appendix says most models were trained for 800 epochs. Also App E2 says Table 7 used 200 epochs, and Table 7 reports lower accuracy for KL loss with alpha=40 than Table 2. So how many epochs were used? 200 is very short training for SimCLR.

* Section 4.2: Please report in the main text the alpha that you used (currently you just say that you "can tune" alpha with KL-MICL to outperform InfoNCE). It's interesting that the value is 40 because it's so large, which makes the loss very different from InfoNCE (with alpha=1).

* In Table 7 you show the results using KL loss and alpha=1 which should be close to SimCLR. You get 77.6% accuracy on CIFAR-10, which is much lower than what one get with SimCLR on CIFAR-10. But that's only using 200 epochs. If you run KL loss with alpha=1 for longer, e.g. for 800 epochs, what performance would you get? Would it be similar to what you report for SimCLR in Table 1 (89.71%)?


MINOR ISSUES:

* Eq 2: is this is a bound of I(X,Y) then maybe it would be clearer if you write it as an inequality, e.g. "I(X,Y) >= ...".

* The very end of section 2: At this point I was wondering if the inequality in Eq (2) becomes an equality if you plug s* from Eq (3) into Eq (2). I think the answer is yes, and I think it can be stated more directly.

* Figure 1: The caption should say that x^g and y^g here refer to positive pairs, i.e. distinct augmentations of the same image. This wasn't immediately clear to me.

* Please add some brief description of how RealNVP works and what it does.

* Table 4: Maybe change "Gaussian" into "f-Gaussian" for clarity?

* Section 4.3: here you change similarity metrics, but it's unclear what loss functions were used? Do you always use the corresponding loss function? So e.g. f-Gaussian similarity for JS loss was used with JS loss, f-Gaussian similarity for Pearson loss was used with Pearson loss etc.? Please clarify.

**Strengths And Weaknesses:**

STRENGTHS:

* The paper suggests a rich family of SSL loss functions, naturally generalizing InfoNCE.
* Comprehensive benchmark on several different datasets.
* Based on the existing theory of f-divergences.

WEAKNESSES:

* Some parts are unclear -- see below.

Overall I think the paper can be accepted to TMLR after some straightforward revisions.

---

> ### Author Response · Authors · 2023-08-03
> **Response to Reviewer DHPy**
>
> We would first like to thank Reviewer DHPy for the prompt reply. We have modified our paper according to your suggestions (marked in the teal color). Here we address your concerns in detail:
>
> ### Major issues
>
> **(1) Relationship between $p+$ and $p(x,y)$.** Sorry for the confusion. Indeed $p+$ and $p(x,y)$ could be used interchangably for the context of contrastive learning, as the positive samples are drawn from the joint distribition. We have added relevant explanation in Section 3.1, right after the definition of $p+$.
>
> **(2) Quality of representation.** Thank you for pointing it out. We followed SimCLR and used a three layer MLP during contrastive learning, but throw it away for linear evaluation. We have added relevant discussion in Section 4.1.  Note that for k-NN evaluation in Table 7 we did not use a projection head.
>
> **(3) Training epochs.** We apologize for the typo, we confirm that for smaller datasets in vision tasks, we train the model for 800 epochs. We have modified it in the main paper. The only execptions are Table 7 and Figure 6, where we use only 200 epochs to demonstrate the choice of $\alpha$ and training stability.
>
> **(4) Mention the $\alpha$ you used.** Thank you for your insightful suggestion. We have added the choice of $\alpha$ in the main paper. Additionally, we want to mention that even when $\alpha=1$, the KL-MICL objective is not identical to InfoNCE (in Section 3.4). Moreover, we are using the $f$-Gaussian similarity function, while InfoNCE applies cosine similarity. Thus, KL-MICL requires special tuning to outperform InfoNCE.
>
> **(5) About $\alpha=1$.** Following the above discussion, in Table 7, we would not expect KL-MICL to perform similarly as SimCLR when $\alpha=1$ because of the above-mentioned differences between the two. Nevertheless, we have performed the same experiments for 800 epochs and observed an accuracy of 83.58% (lower than SimCLR). This observation further provides emprical evidence to our claim above: KL-MICL is very different from InfoNCE and needs special tuning on $\alpha$ to perform well.
>
> ### Minor Issues
>
> **(1) Eq (2).** Thank you. We have rewritten Eq 2.
>
> **(2) The very end of section 2:** Indeed we obtain equality if one plugs $s^*(x,y)$ into Eq (2). We have added this explicitly at the end of Section 2.
>
> **(3) Figure 1.** Thank you, we have added the definitions in the caption.
>
> **(4) Please add some brief description of how RealNVP works and what it does.** We have added the description on RealNVP on Page 5 (footnote).
>
> **(5)Table 4.** We have changed "Gaussian" to "$f$-Gaussian" in Table 4.
>
> **(6)Section 4.3.** We always use the corresponding loss function. We have added the clarification in Section 4.3.

---

> > ### Comment · Reviewer_DHPy · 2023-08-04
> > **Thank you**
> >
> > Thank you for the quick and helpful revision.
> >
> > Some things here are still not entirely clear to me (which I am sure is due to my confusion):
> >
> > > Indeed p+ and p(x,y) could be used interchangably for the context of contrastive learning, as the positive samples are drawn from the joint distribition. We have added relevant explanation in Section 3.1
> >
> > You added this statement, but not really an _explanation_... Could you explain it more in detail? Why does it make sense to think of p+ as the joint distribution p(x,y)? I must be missing something here, from the very beginning. Does this assume that x and y are positively related? Then of course I can see that p+ = p(x,y)... But you don't write it like that, you just say "We denote p+ as the distribution of positive pairs, i.e., two samples with similar feature embeddings (joint distribution)". I think this part can be elaborated on.
> >
> > > Additionally, we want to mention that even when , the KL-MICL objective is not identical to InfoNCE (in Section 3.4).
> >
> > This is another part that is not very clear to me. Is it not possible to write InfoNCE as a f-MICL objective? If not, why not? You begin the paper by saying that you standard InfoNCE can be seen as KL and that you want to "generalize" the KL to other f-divergences. If so, how come your generalization cannot describe InfoNCE itself? Maybe a little more discussion of this point would be helpful.
> >
> > > Moreover, we are using the f-Gaussian similarity function, while InfoNCE applies cosine similarity.
> >
> > This made me wonder, how different is f-Gaussian similarity for KL loss (on the S^{d-1} sphere) from the cosine similarity. You show in Table 4 that they perform slightly differently, but it could be interesting to write down the expression for the f-Gaussian similarity in this case, to see how different it is from the cosine.
> >
> > > Nevertheless, we have performed the same experiments for 800 epochs and observed an accuracy of 83.58% (lower than SimCLR). This observation further provides emprical evidence to our claim above: KL-MICL is very different from InfoNCE and needs special tuning on to perform well.
> >
> > Thanks for doing this experiment. Could be helpful to add this result to the supplementary information!

---

> > > ### Author Response · Authors · 2023-08-05
> > > **Further Clarification**
> > >
> > > Thank you for the quick feedback! We understand some of the confusion still remain and we would like to elaborate more to clarify them:
> > >
> > > > Why does it make sense to think of p+ as the joint distribution p(x,y)?
> > >
> > > (1) In contrastive learning, to construct positive pairs, we always apply different data augmentations on the same image, thus $x$ and $y$ are positively related (e.g., the upper part of Figure 2);
> > > (2) We realize it is not immediately clear at the beginning of Section 3. Thus we change it to "We denote $p_+$ as the distribution of positive pairs, i.e., two samples with similar feature embeddings (joint distribution, e.g., **the same image with different data augmentation**)", as a concrete example in computer vision.
> > >
> > > >  Is it not possible to write InfoNCE as a f-MICL objective?
> > >
> > > (1) Firstly, we want to emphasize that InfoNCE cannot be seen as estimating the KL-based mutual information *directly*, but **can be interpreted as a lower bound of the mutual information**. We were very careful regarding the precision of this claim in our abstract and introduction.
> > >
> > > (2) Secondly, in our paper, we formulate the objective as *directly* estimating $f$-mutual information from its definition. Due to this difference, we **cannot** write InfoNCE naively as a $f$-MICL objective.
> > >
> > > (3) Thirdly, we can indeed describe InfoNCE using our framework: in Section 3.4, we find that KL-MICL is equivalent to a Donsker-Varadha shift transformation of InfoNCE, thus InfoNCE can be understood as an upper bound of KL-MICL.
> > >
> > > (4) In summary, we characterize our generalization in Figure 3: where we identify some InfoNCE-based objectives: AU objective and Spectral Contrastive loss are special cases of $f$-MICL while InfoNCE itself can be transformed to KL-MICL.
> > >
> > > > how different is f-Gaussian similarity for KL loss (on the $S^{d-1}$ sphere) from the cosine similarity?
> > >
> > > Under Eq (14) on page 7, we have shown the equivalence for KL loss:
> > >
> > > "Note that for KL, the Gaussian similarity is equivalent to the cosine similarity with a scaling factor: $-\|x^g - y^g\|^2=2(x^g)^{\top}y^g-2$."
> > >
> > > In the implementation, if we apply cosine similarity with the correct scaling factor, it would recover the Gaussian similarity exactly.
> > >
> > > > Could be helpful to add this result to the supplementary information
> > >
> > > Thank you, we have added the results to the appendix.
> > >
> > >
> > > Thank you again for your constructive suggestions and please let us know if your concerns have been addressed.

---

> > > > ### Comment · Reviewer_DHPy · 2023-08-05
> > > > **Thank you**
> > > >
> > > > Thanks for you reply. I don't have any more comments and recommend acceptance.
> > > >
> > > > (This is the first time I review for TMLR so I am not very familiar with the process. So far I did not see any other reviews. Usually when reviewing for a ML conference or a journal, a reviewer gets to see other reviews together with the authors' rebuttal. So I just want to make it clear that I have not seen any other reviews yet.)

---

### Review · Reviewer_NShA · 2023-08-18

**Summary Of Contributions:**

The paper proposes $f$-MICL, a framework for contrastive learning. It generalizes InfoNCE by using $f$-divergence instead of KL divergence. This provides a wide choice of objectives as we can use different $f$-divergences, and it is shown that several existing contrastive losses can be interpreted as a specialization of the $f$-MICL by choosing a specific $f$-divergence and proper hyperparameters. The framework also provides theoretical guidance to choosing the optimal similarity function. Experiments show that $f$-MICL achieves comparable or better linear evaluation accuracy across multiple datasets compared to several popular self-supervised learning methods. It is also shown that the proposed $f$-Gaussian similarity works better than cosine similarity.

**Audience:**

Yes

**Claims And Evidence:**

Yes

**Requested Changes:**

Overall I think the proposed framework makes sense. There are some clarity issues that I have listed in Weaknesses, and I hope the authors could address them, especially the second and third point.

**Strengths And Weaknesses:**

- Strengths
    - Experiments are quite extensive, covering both vision and language datasets and several popular baselines. Results are averaged over 3 seeds.
    - The paper also compares different choices of the $f$-divergence and similarity function.
    - The paper provides empirical support for its assumption (Fig 1)
    - The paper is generally well written.
- Weaknesses
    - I thought there should be $\text{sup}_{s \in \mathcal{F}}$ in Eq 4. To approximate this $\text{sup}$, we use $s = s_f$ defined in Eq 6.
    - Fig 1 only shows that Assumption 3 approximately holds for a specific feature encoder $g$ (that is trained by SimCLR loss). However, the paper seems to suggest that Assumption 3 should hold for arbitrary $g$.
    - I don't see how Eq 6 is derived from Eq 5 and Assumption 3.
    - I'm wondering why the negative samples should use the same data augmentation (Eq 8). Can they use different data augmentations?
    - What is "max" in Fig 2?
    - The $\alpha$ values used in vision ($\alpha = 40$) and language ($\alpha = 409600$) tasks are drastically different. Can you provide some guidelines on how to choose $\alpha$?
    - Some text are colored and I don't see why. Please fix that.

---

> ### Author Response · Authors · 2023-08-21
> **Response to Reviewer NShA**
>
> Thank you for your review and great questions, we have modified our paper accordingly where the relevant changes are marked in violet. Here we provide clarifications.
>
> **(1) Equation (4)**: indeed we aim at maximizing the objective function in Equation (4) and there should be a $\text{sup}_{s \in \mathcal{F}}$, we have rewritten Equation (4) in our revised draft.
>
> **(2) Figure 1:** in Appendix D.3 (Figure 7), we provide additional results on KL-MICL, which verifies Assumption 3 as well. Of course, we do not claim that it holds true for an arbitrary encoder $g$, but for the context of $f$-MICL, we verify that the $f$-Gaussian kernel is indeed preferable in Table 4.
>
> **(3) Equation (6):** Sorry for the confusion here, in Equation (6), we directly substitute $p_g(x^g,y^g)$ with our $f$-Gaussian kernel. As for the product of the marginals $p_g(x^g)p_g(y^g)$, we choose $x^g$ and $y^g$ to be both normalized such that when we marginalize either of them, the integral would be a constant due to rotational invariance. As a result, the product of marginals $p_g(x^g)p_g(y^g)$ is also a constant and can be thus absorbed into a weighting parameter $\alpha$ that we introduce in Section 3.3. We have added Lemma 4 and its proof in Appendix A.3 in the revision.
>
> **(4) Augmentation:** Sorry for the typo here that causes the confusion: indeed it does not have to be the same augmentation, and in our implementation, we draw the augmentations independently for the anchor and the negative samples.
>
> **(5) max** denotes maximization, we have made them clear in the updated draft.
>
> **(6) guidance for the choice of $\alpha$:** in Appendix D.2, we apply a grid search for $\alpha$ from $1$ to $10^6$, where we examine the loss after training for 1 epoch until we find the best $\alpha$ that induces the most loss drop. In general, $\alpha$ balances between the positive and negative terms. We leave the theoretical study of $\alpha$ to future work.
>
>
> **(7) text in other color:** we apologize that the colored text was to indicate the changes made when addressing Reviewer DHPy's concerns. We have changed them back to black.

---

### Review · Reviewer_zeWK · 2023-08-30

**Summary Of Contributions:**

The paper studied a new class of contrastive representation learning objective called f-MICL. f-MICL replaces the KL divergence in Shannon mutual information with the f-divergence. The representation learning framework adopted in the paper is the same as InfoNCE except that mutual information is generalized to f-MICL. In addition, the author proposed the f-Gaussian similarity function by assuming that the joint feature distribution is proportional to a Gaussian. Experiments on vision and text datasets demonstrated the benefit of f-Gaussian similarity over the cosine similarity.

**Audience:**

Yes

**Broader Impact Concerns:**

No concern

**Claims And Evidence:**

No

**Requested Changes:**

1. The author needs to emphasize its novelty over Deep InfoMax, and compare f-Gaussian with regularizing the marginal distribution with a compact prior distribution.
2. I think the author needs to revise Figure 1 or find other ways to justify the validity of assuming the marginal feature distribution be a Gaussian.
3. The author needs to discuss how to choose the best f empirically.

**Strengths And Weaknesses:**

Strengths
1. f-MICL is a generalization of Shannon mutual information. According to Table 2, the KL-based mutual information does not always perform the best and may underperform JS/Pearson/VLC-based mutual information. Thus, f-MICL offers more options to practitioners and people can search for the most appropriate loss function for their downstream application.
2. Experiment results show that f-Gaussian similarity is beneficial.

Weaknesses
1. The paper is highly relevant to Deep InfoMax (https://arxiv.org/pdf/1808.06670.pdf). For example, Deep InfoMax (DIM) proposed to control the characteristics of the representation by matching the push-forward marginal with a prior distribution (DIM used the uniform distribution in the experiment and mentioned that Gaussian distribution under-performs uniform as a prior). The f-Gaussian idea proposed by the author is conceptually similar to the idea in DIM. By adopting the f-Gaussian similarity function, the author intrinsically assumes that the marginal distribution is a Gaussian. This also encourages the compactness of the learned representation like the regularization term in DIM. Moreover, DIM generalizes KL-based MI to JS-based MI via the technique in the f-GAN paper. The idea of extending MI to f-MICL is very incremental since prior work has already tried to extend MI via f-divergence.
2. Although the KL-based MI does not always perform the best, no f-divergence clearly wins over the others according to Table 2 and Table 3. The author mentioned that "We will discuss how to choose the best f empirically in §4." but I haven't seen the algorithm of choosing the best f for a given dataset. In real-world applications, trying out all variants can be computationally impractical.
3. Adopting RealNVP for verifying the Gaussian assumption in Figure 1 is questionable since RealNVP usually assumes the latent distribution follow a Gaussian.

---

> ### Author Response · Authors · 2023-09-04
> **Response to Reviewer zeWK**
>
> Thank you for your informative review! We have modified our paper accordingly, with the relevant changes marked in blue. Here we address your concerns in detail:
>
> >**(1) Our paper is highly relevant to Deep InfoMax:**
>
> Thank you for mentioning this reference. We agree that our paper is relevant to Deep InfoMax, which we compared with in Section 5, paragraph "Mutual Information." However, our paper is still different from DIM, and here we emphasize some key points:
>
> Let's first write down the objectives of $f$-MICL and DIM using our notations:
>   - $\mathcal\{L\}_{f-\text{MICL}}=\mathbb{E} _{(x,y)\sim p _+}s_f(x^g,y^g)-\mathbb{E} _{(x,y)\sim p _{\times}}f^*\circ s_f(x^g,y^g)$
>   - $\mathcal{L}_{\text{DIM}} = \mathbb{E} _{x\sim p}s_f(x,x^g)- \mathbb{E} _{(x,y)\sim p _{\times}}f^*\circ s_f(y,x^g)$
>
> Note that here we write $\mathcal{L}_{\text{DIM}}$ for general $f$-divergences while the paper only considers KL and JS divergences.
>
> Now we are ready to note some key differences and our advantages:
>
> - **Different tasks**: We consider the *contrastive learning* task, where we aim at maximizing the mutual information between two views of samples $x^g$ and $y^g$ drawn from the joint distribution, while DIM considers *deep representation learning*, where maximizing the mutual information between input data $x$ and the learned representation $x^g$ is desirable.
> - **Generality:** While DIM only introduces the objective for KL and JS divergence, we provide explicit design recipes for various $f$-divergences.
> - **Understanding different contrastive learning methods:** As our $f$-MICL objective is designed specifically for contrastive learning, we can use our framework to understand and generalize other contrastive learning methods, detailed in Section 3.4.
> - **Empirical Superiority:** We believe our study on contrastive learning is valuable as it performs much better than DIM-based objectives in practice. For example, we achieve 90.61% for KL-MICL on CIFAR-10 while DIM only reaches 75.57% in the best scenario.
>
> To clearly show the differences between our method and other representation learning methods that apply $f$-divergences, we have added Table 5 in Section 5 for a detailed comparison regarding different aspects of various methods.
>
>
> >**(2) Our $f$-Gaussian similarity is similar to DIM**:
>
> We believe there are some misunderstandings with respect to our $f$-Gaussian similarity, which we clarify below:
> - Using $f$-Gaussain similarity we study the optimal *similarity function $s(\cdot)$*, whereas DIM chooses to parameterize $s(\cdot)$ with a neural network. We indeed implemented the latter practice for the design of $s(\cdot)$, and found such a parameterization did not work in the context of contrastive learning: on the CIFAR-10 dataset, we can only achieve around 70\% accuracy in the best scenario.
> - The reviewer mentioned that "DIM proposed to control the characteristics of the representation by matching the push-forward marginal with a prior distribution." We note that such a matching process is to regularize the **feature representations of all samples,** in contrast to our work (see the next point).
> - For $f$-MICL,  such adversarial regularization is not required: we demonstrate that by minimizing the second term of our objective function (in Equation (8)), our feature representations automatically exhibit the uniformity property, as shown in Theorem 5 and our experiments in Section 4.4, which also matches the discovery of DIM. We note that uniformity is another advantage of $f$-MICL over DIM.
> - Moreover, the reviewer mentioned that "the author intrinsically assumes that the marginal distribution is a Gaussian." We want to clarify that this is *not true*, as we only assume the **joint feature distribution** is proportional to a Gaussian kernel in Assumption 3, which is empirically verified in Figure 1.
>
>
> >**(3) The idea of extending MI to f-MICL is very incremental since prior work has already tried to extend MI via f-divergence:**
>
> - We agree that extending MI to $f$-MI is not a contribution of ours, and we attributed it in Definition 1 (Section 2) to Csiszár (1967).
> - Although $f$-divergences have been widely used in other applications (which we discussed in Section 5), we believe extending $f$-MI to contrastive learning and constructing $f$-MICL is non-trivial as the general framework:
>     - (a) provides new insights on understanding existing methods;
>     - (b) enables us to design the $f$-Gaussian similarity function;
>     - (c) improves the performance of existing contrastive learning methods by only varying the objective function without any architectural design.

---

> > ### Author Response · Authors · 2023-09-04
> > **Response to Reviewer zeWK (continued)**
> >
> > > **(4) Adopting RealNVP for verifying the Gaussian assumption in Figure 1 is questionable since RealNVP usually assumes the latent distribution follows a Gaussian.**
> >
> > We understand the confusion and here we want to provide more details to justify our choice of RealNVP:
> > - To train a RealNVP model for estimating the log density of the joint distribution $\log p_g(x^g,y^g)$, we take:
> >   - (a) the training data $x$, $y$, which are  different data augmentations on the same image sample;
> >   - (b) a pre-trained feature extractor $g$ (learned by SimCLR in Figure 1) and acquire the feature representations $x^g$, $y^g$, which serves as **input** to the RealNVP algorithm.
> > - The RealNVP learns an invertible transformation $f$ from the input space ($x^g$, $y^g$) to the latent space $z$, where $z=f(x^g,y^g)$. Note that here **$z$ is a prior distribution, which we choose to be Gaussian, but any continuous distribution on $z$ would work since it only requires changing $f$ accordingly**.
> > - Thus, the assumption of Gaussian distribution only lies on the prior distribution of $z$, and we do not pose any assumption on the input space $(x^g, y^g)$. Therefore, the learned log density of the joint distribution has no assumption on Gaussian as well. To this end, we hope that we have shown that our validation on Assumption 3 and Figure 1 are indeed valid.
> >
> > To further confirm the choice of the prior distribution on $z$ does not affect our validation of Assumption 3, we further change the prior distribution to uniform and retrain the RealNVP model. The updated Figure 1 and the new Figure 8 (where we conduct 5 different combinations of random data augmentations ) shows that a different prior distribution does not affect the estimation of the joint distribution, and the linear relationship holds approximately for the uniform prior as well.
> >
> >
> >
> >
> > > **(5) The author needs to discuss how to choose the best f empirically.**
> >
> >
> > - As we mentioned in Section 4.2 (third paragraph) and Section 6 (limitations and future work ),  it is yet unclear how to choose an optimal f based on a task and a dataset in theory, such that we usually rely on a validation set in practice for selection.
> > - Of course, the ideal way to choose the best $f$ is to learn an optimal $f$-divergence using a parameterized neural network, which we leave as future work in Section 6.
> > - The reviewer mentions that "In real-world applications, trying out all variants can be computationally impractical." However, in practice, we find such a searching process for $f$ can be easily executed:
> >   - (a) using our theory, we can already rule out several non-satisfying $f$-divergences;
> >   - (b) our experiments generally confirm that only a few choices of $f$ yield competitive performance (i,e., KL, JS, Pearson, VLC) for popular vision and language tasks;
> >   - (c) we find that these $f$-MICL objectives share the same hyperparameters in Table 7 (Appendix D);
> >   - (d) as a result, searching optimal $f$ is no harder than performing a hyperparameter search, where the repetition cost is only linear (4 in our case).

---

### Decision · Action_Editors · 2023-09-20

**Recommendation:** Accept with minor revision

**Comment:**

This paper considered a generalized view of contrastive learning by extending the KL divergence and dot product kernel to a generalized setting (f-divergence and a general kernel view). Most reviewers appreciated the insights and results and recommended acceptance. However, there are some concerns raised by reviewers. After reviewing the paper and the concerns, I think the current version can be accepted if some very minor points are clarified. Below are detailed thoughts on the concerns:

- [About novelty]. *Generalizing MI to f-MI is incremental since it has been widely adopted in previous papers...* I would think this is fine since the main scope of TMLR is to accept papers with certain interesting insights. I would agree that the extension is interesting because it provides a general view for understanding contrastive learning.

- [About f-Gaussian similarity function] I agree that the f-Gaussian similarity seems relatively strong, while it really depends on the *data assumption*. Based on Fig. 1, the current approximation is empirically validated on the benchmark dataset. I would suggest that  authors discuss this limitation.

Other suggestions. There is no need to fix them in the current version, but it would be great to have a short discussion as a limitation/future work.

- On the inherent limitation of variational terms of f-divergence. [1] pointed out the statistical limitation of using variational forms to measure mutual information. I would think that there might be similar limitations when using the duality term of f-divergence. It might be an open problem and interesting to explore.

- Copula in joint distribution modeling. Concerns about f-Gaussian similarity function are quite related to Copula [2] by forming different joint distributions by giving marginal distributions. I would think that the current approximation could be considered as a specific setting under Gaussian Copula. Moreover the positive and negative pair have specific dependence relations, it might be quite interesting to explore a general form.

Based on these, I would recommend an acceptance.

References

[1] Formal Limitations on the Measurement of Mutual Information. AISTAS 2019

[2] Introduction to Copula https://bochang.me/blog/posts/copula/

**Audience:**

Yes

**Claims And Evidence:**

Yes. Some minor points need clarifications.

---

> ### Author Response · Authors · 2023-10-25
> **Response to Action Editors**
>
> We would like to thank the action editor for acknowledging our contribution and providing valuable suggestions. We have carefully addressed the concerns in the camera-ready version and we summarize the changes below:
>
> - [About f-Gaussian similarity function]: we have added a footnote on Page 6 discussing the limitation of our verification in Figure 1:
>
>   >The linear relationship in Figure 1 might also depend on the data, i.e., the CIFAR-10 dataset here. In practice, other customized datasets might require additional verification.
>
> - [Inherent limitation of variational terms of f-divergence]: thank you for providing this reference, which we have cited and discussed in possible future work in Section 6.
>
> - [Copula in joint distribution modeling]: thank you for the great insights! Based on your suggestions, we have rewritten the paragraph below Assumption 3 (on Page 5) a bit and pointed out the connections to the copula in footnote 2.